# SafeDiffuser: Safe Planning with Diffusion Probabilistic Models

**Wei Xiao**[1]*, **Tsun-Hsuan Wang**[1], **Chuang Gan**[2], **Ramin Hasani**[1], **Mathias Lechner**[1],
**Daniela Rus**[1]

[1] Computer Science and Artificial Intelligence Lab, MIT, Cambridge, MA 02139, USA
[2] UMass Amherst and MIT-IBM Watson AI Lab, USA
`weixy@mit.edu`

## Abstract

Diffusion models have shown promise in data-driven planning. While these planners are commonly employed in applications where decisions are critical, they still lack established safety guarantees. In this paper, we address this limitation by introducing SafeDiffuser, a method to equip diffusion models with safety guarantees via control barrier functions. The key idea of our approach is to embed finite-time diffusion invariance, i.e., a form of specification consisting of safety constraints, into the denoising diffusion procedure. This way we enable data generation under safety constraints. We show that SafeDiffusers maintain the generative performance of diffusion models while also providing robustness in safe data generation. We evaluate our method on a series of tasks, including maze path generation, legged robot locomotion, and 3D space manipulation, and demonstrate the advantages of robustness over vanilla diffusion models[1].

## 1 Introduction

Diffusion models Sohl-Dickstein et al. (2015) Ho et al. (2020) are a family of generative modeling approaches that have enabled major breakthroughs in image synthesis Dhariwal & Nichol (2021) Du et al. (2020b) Saharia et al. (2022). Recently, diffusion models, termed diffusers Janner et al. (2022), have shown promise in trajectory planning for a variety of

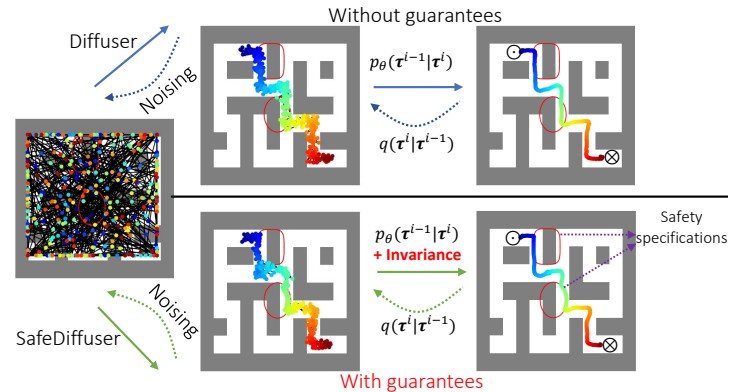

Figure 1: Our proposed SafeDiffuser (lower) generates safe trajectories with guarantees, while the diffuser (upper) fails (from $\odot$ to $\otimes$).

robotic tasks. Compared to existing planning methods, diffusion models (a) enable long-horizon planning with multi-modal action distributions and stable training, (b) easily scale to high-dimensional trajectory planning, and (c) offer flexiblity for behavior synthesis.

During inference, the diffuser, conditioned on the current state and objectives, begins with Gaussian noise to generate clean planning trajectories. From these, a control policy is derived. After applying this control policy for one step forward, a new state is obtained, and the diffusion procedure is rerun to generate a new planning trajectory. This process repeats until the objective is achieved.

Although these planners are primarily applied in safety-critical applications, no known safety guarantees have been established for them. For instance, the planning trajectory could easily violate safety constraints in the maze (as shown in Fig. 1). This shortcoming necessitates a fundamental im-

---

*Corresponding to: Wei Xiao
[1]Videos and code are available at: `https://safediffuser.github.io/safediffuser/`

provement to diffusion models, ensuring the safe generation of planning trajectories in safety-critical applications, such as trustworthy policy learning Xiao et al. (2023a).

In this paper, we propose to equip diffusion models with specification guarantees using finite-time diffusion invariance (i.e., safety satisfaction within finite diffusion time for all planning times). An invariant set is a form of specification primarily consisting of safety constraints in planning tasks. We ensure that diffusion models are invariant to uncertainties concerning safety during the diffusion procedure. We achieve safety by combining receding horizon control (RHC) with diffusion models. In RHC, we compute safe paths incrementally. The key insight in this work is to replace traditional planning with diffusion-based path generation, allowing a broader exploration of the path space and simplifies the incorporation of additional constraints. The computed path is integrated with simulation to ensure it can be safely executed.

To equip diffusers with specifications guarantees, we first find diffusion dynamics for the denoising diffusion procedure. Then, we use control barrier functions (CBFs) Ames et al. (2017) Glotfelter et al. (2017) Nguyen & Sreenath (2016) Xiao & Belta (2019), to formally guarantee the satisfaction of specifications. CBFs work well in planning time using robot dynamics. However, applying CBFs to diffusion models poses extra challenges since the generated data is not directly associated with robot dynamics. This makes the use of CBFs non-trivial. In contrast to existing literature, (i) we suggest embedding invariance directly into the diffusion time for diffusers. Thus, finite-time invariance is required in diffusers since specifications are usually violated as the trajectory is initially Gaussian noise. (ii) We propose to add diffusion time components in invariance to address local trap problems (i.e., trajectory points that get stuck at obstacle boundaries) that are prominent in planning. (iii) We present an optimization approach to incorporate invariance into diffusion to maximally preserve the performance.

This paper contributes the following:
- We introduce formal guarantees for diffusion probabilistic models via control-theoretic invariance.
- We present a novel notion of finite-time diffusion invariance, as well as using a class of CBFs to incorporate it into the diffusion. We propose three different safe diffusers, and show how we may address the local trap problem from specifications that are prominent in planning tasks.
- We demonstrate the effectiveness of our method on a variety of planning tasks using diffusion, including safe planning in maze, robot locomotion, and manipulation.

## 2 PRELIMINARIES

In this section, we provide background on diffusion models and forward invariance in control theory.

**Diffusion Probabilistic Models.** Diffusion probabilistic models Sohl-Dickstein et al. (2015); Ho et al. (2020); Janner et al. (2022) are a type of latent variable models. They describe the process of data generation as a series of iterative denoising steps. Here, the model is represented as $p_\theta(\tau^{i-1}|\tau^i)$, $i \in \{1, \ldots, N\}$, where $\tau^1, \ldots, \tau^N$ are latent variables mirroring the dimension of the original, noise-free data $\tau^0 \sim q(\tau^0)$, and $N$ signifies the total number of denoising steps. This denoising sequence is essentially the inverse of a forward diffusion process denoted as $q(\tau^i|\tau^{i-1})$ where the initial clean data is progressively degraded by adding noise. The process of generating data through denoising is expressed as Janner et al. (2022):

$$p_\theta(\tau^0) = \int p_\theta(\tau^{0:N}) d\tau^{1:N} = \int p(\tau^N) \prod_{i=1}^{N} p_\theta(\tau^{i-1}|\tau^i) d\tau^{1:N}. \tag{1}$$

In this equation, $p(\tau^N)$ represents a standard Gaussian prior distribution. The joint distribution $p_\theta(\tau^{0:N})$ is defined as a Markov chain with learned Gaussian transitions that commence at $p(\tau^N)$ Janner et al. (2022). The optimization parameter $\theta$ is achieved by minimizing the common variational bound on the negative log-likelihood of the reverse process, formalized as Janner et al. (2022): $\theta^* = \arg\min_\theta \mathbb{E}_{\tau^0}\left[-\log p_\theta(\tau^0)\right]$. The forward diffusion process, denoted as $q(\tau^i|\tau^{i-1})$, is typically predefined. Conversely, the reverse process is frequently characterized as a Gaussian process, featuring a mean and variance that vary depending on time.

**Notations.** For the sake of consistency, we keep our notations as that proposed in Janner et al. (2022) as follows: Here, two distinct 'times' are discussed: one associated with the diffusion process and

the other with the planning horizon. These are differentiated as follows: superscripts (employing $i$ when unspecified) indicate the diffusion time of a trajectory or state, whereas subscripts (using $k$ when unspecified) denote the planning time of a state within the trajectory. For instance, $\boldsymbol{\tau}^0$ refers to the trajectory at the initial denoising diffusion time step, which is a noiseless trajectory. In a similar vein, $\boldsymbol{\tau}_k^0$ represents the state at the $k^{th}$ planning time step during the first denoising diffusion step, indicating a noiseless state. When clarity permits, we simplify this notation to $\boldsymbol{\tau}_k = \boldsymbol{\tau}_k^0$ (and similarly $\boldsymbol{\tau} = \boldsymbol{\tau}^0$). Moreover, a trajectory $\boldsymbol{\tau}^i$ is conceptualized as a sequence of states across planning time, articulated as $\boldsymbol{\tau}^i = (\boldsymbol{\tau}_0^i, \boldsymbol{\tau}_1^i, \ldots, \boldsymbol{\tau}_k^i, \ldots, \boldsymbol{\tau}_H^i)$, where $H \in \mathbb{N}$ defines the planning horizon.

**Forward Invariance in Control Theory.** We consider an affine control system:

$$\dot{\boldsymbol{x}}_t = f(\boldsymbol{x}_t) + g(\boldsymbol{x}_t)\boldsymbol{u}_t, \tag{2}$$

where $\boldsymbol{x}_t \in \mathbb{R}^n$, $f : \mathbb{R}^n \to \mathbb{R}^n$ and $g : \mathbb{R}^n \to \mathbb{R}^{n \times q}$ are locally Lipschitz, and $\boldsymbol{u}_t \in U \subset \mathbb{R}^q$, where $U$ denotes a control constraint set. $\dot{\boldsymbol{x}}_t$ denotes the (planning) time derivative.

Consider a safety specification $b(\boldsymbol{x}_t) \geq 0$ for (2), where $b : \mathbb{R}^n \to \mathbb{R}$ is continuously differentiable, we define a safe set: $C := \{\boldsymbol{x}_t \in \mathbb{R}^n : b(\boldsymbol{x}_t) \geq 0\}$.

**Definition 2.1.** *(**Control Barrier Function (CBF)** Ames et al. (2017)): A function $b : \mathbb{R}^n \to \mathbb{R}$ is a CBF if there exists an extended class $\mathcal{K}$ function $\alpha$ (strictly increasing and passing the origin) s.t.*

$$\sup_{\boldsymbol{u}_t \in U} [L_f b(\boldsymbol{x}_t) + [L_g b(\boldsymbol{x}_t)]\boldsymbol{u}_t + \alpha(b(\boldsymbol{x}_t))] \geq 0, \tag{3}$$

*for all $\boldsymbol{x}_t \in C$. Where $L_f b(x_t) = \frac{db(x_t)}{dx_t} f(x_t)$ and $L_g b(x_t) = \frac{db(x_t)}{dx_t} g(x_t)$.*

**Theorem 2.2** (Ames et al. (2017))**.** *Given a CBF $b(\boldsymbol{x}_t)$ as in Def. 2.1, if $\boldsymbol{x}_0 \in C$, then any Lipschitz continuous controller $\boldsymbol{u}_t$ that satisfies (3), $\forall t \geq 0$ renders $C$ forward invariant for (2), i.e. $b(x_t) \geq 0, \forall t$.*

If we need to differentiate $b(\boldsymbol{x}_t)$ more than once along the dynamics (2) until the control $\boldsymbol{u}_t$ explicitly shows, we use a high-order CBF Nguyen & Sreenath (2016) Xiao & Belta (2019) as a general form of CBF to guarantee safety for (2). CBFs are usually used to transform nonlinear optimal control problems into convex optimizations. Time is usually discretized, and the inter-sampling effect is considered Ames et al. (2017). This discretization method matches with the diffusion procedure in which we have to generate data within each diffusion (discretized) time step. In this work, *we map the forward invariance in control theory to finite time diffusion invariance in diffusion models*, where we incorporate CBFs into the diffusion time $\cdot^i$ as opposed to their regular applications in planning time $\cdot_k$. In addition, we show how we may address the local traps (i.e., trajectory points getting stuck at obstacle boundaries) during diffusion.

## 3 SAFE DIFFUSER

In this section, we propose three different safe diffusers to ensure the safe generation of data in diffusion, i.e., to ensure the satisfaction of specifications $b(\boldsymbol{\tau}_k) \geq 0, \forall k \in \{0, \ldots, H\}$. Each of the proposed safe diffusers has its own flexibility, such as avoiding local traps in planning. We consider discretized system states in the sequel. Safety in continuous planning time can be guaranteed using a lower hierarchical control framework employing other CBFs, as in Ames et al. (2017); Nguyen & Sreenath (2016); Xiao & Belta (2019).

In the denoising diffusion procedure, since the learned Gaussian transitions start at $p(\boldsymbol{\tau}^N) \sim \mathcal{N}(0, \boldsymbol{I})$, it is highly likely that specifications are initially violated, i.e., $\exists k \in \{0, \ldots, H\}, b(\boldsymbol{\tau}_k^N) < 0$. For safe data generation, we wish to have $b(\boldsymbol{\tau}_k^0) \geq 0 (i.e., b(\boldsymbol{\tau}_k) \geq 0), \forall k \in \{0, \ldots, H\}$. Since the maximum denoising diffusion step $N$ is limited, this needs to be guaranteed in a finite diffusion time step. Therefore, we propose the finite-time diffusion invariance of the diffusion procedure as follows:

**Definition 3.1.** *(Finite-time Diffusion Invariance) If there exists $i \in \{0, \ldots, N\}$ such that $b(\boldsymbol{\tau}_k^j) \geq 0, \forall k \in \{0, \ldots, H\}, \forall j \leq i$, then a denoising diffusion procedure $p_\theta(\boldsymbol{\tau}^{i-1}|\boldsymbol{\tau}^i), i \in \{1, \ldots, N\}$ with respect to a specification $b(\boldsymbol{\tau}_k) \geq 0, \forall k \in \{0, \ldots, H\}$ is finite-time diffusion invariant.*

The above definition can be interpreted as that if $b(\boldsymbol{\tau}_k^N) \geq 0, k \in \{0, \ldots, H\}$ (i.e., initial condition is within the safe set), then we require $b(\boldsymbol{\tau}_k^i) \geq 0, \forall i \in \{0, \ldots, N\}$ (i.e., the system is always in the

safe set; similar to the forward invariance in Thm. 2.2); otherwise, we require that $b(\tau_k^j) \geq 0, \forall j \in \{0, \ldots, i\}, i \in \{0, \ldots, N\}$, where $i$ is a finite diffusion time, (i.e., the system converges to the safe set). The finite-time diffusion invariance implies safety of generated outputs from diffusion models.

We also formally define the local trap problem for navigation planning problems as follows:

**Definition 3.2.** *(Local trap) A local trap problem happens for the denoising diffusion procedure If there exists $k \in \{0, \ldots, H\}$ such that $b(\tau_k) = 0$ and $||\tau_k - \tau_{k-1}|| > \delta$, where $\delta > 0$ is some threshold.*

In the following, we propose three methods for finite-time diffusion invariance. The first method is a general form of the safe-diffuser (Sec. 3.1), and the other two (Sec. 3.2-3.3 are variants to address local traps in planning. We show their comparisons in Table 1 and the choice principles at the end.

The safe denoising diffusion procedure is considered at every diffusion step. Following (1), a sample $\tau^j, j \in \{0, \ldots, N-1\}$ follows the data distribution at the diffusion time $j \in \{0, \ldots, N-1\}$, and it is given by:

$$\tau^j \sim p_\theta(\tau^j) = \int p(\tau^N) \prod_{i=j+1}^{N} p_\theta(\tau^{i-1}|\tau^i) d\tau^{j+1:N}. \tag{4}$$

The denoising diffusion dynamics are then given by:

Table 1: Comparison of RoS, ReS, and TVS diffusers. Items in the first row are short for Local-Trap Free (LTF), safety enforcing approach (APPROACH), Dimension of Decision variable (DD), Hyper-Parameter Free (HPF).

| METHOD | LTF | APPROACH | DD | HPF |
|---|---|---|---|---|
| RoS | × | ROBUST | H | √ |
| ReS | √ | ROBUST | 2H | × |
| TVS | √ | TIME FUNCTION | H | × |

$$\dot{\tau}^j = \lim_{\Delta\tau \to 0} \frac{\tau^j - \tau^{j+1}}{\Delta\tau}, \tag{5}$$

where $\dot{\tau}$ is the (diffusion) time derivative of $\tau$. $\Delta\tau > 0$ is a small enough diffusion time step length during implementations, and $\tau^{j+1}$ is available from the last diffusion step. Notably, we enforce invariance along diffusion dynamics instead of robot dynamics (commonly in control theory). The diffusion models (1) could be in discrete Janner et al. (2022) or continuous time (such as those based on stochastic differentiable equations Song et al. (2020)). If the models are in discrete time, as the outputs of the model are bounded, the above diffusion dynamics are Lipschitz; Otherwise, the above model is also practically Lipschitz as long as the activation functions are Lipschitz (the stochastic components are practically sampled from bounded distributions). Further, the Lipschitz constant can be reduced using the sharing conditions Yang et al. (2023) in the diffusion time interval with large Lipschitz constants.

In order to impose finite-time diffusion invariance on the diffusion procedure, we wish to make diffusion dynamics (5) controllable. We reformulate (5) as

$$\dot{\tau}^j = \lim_{\Delta\tau \to 0} \frac{\tau^j - \tau^{j+1}}{\Delta\tau} + \Delta\nu^j := \nu^j, \tag{6}$$

where $\Delta\nu^j$ is a perturbation to the diffusion procedure in order to make the generated trajectory safe. The above equation corresponds to the system dynamics (2) ($\Delta\nu^j$ is the corresponding control). On the other hand, we wish to make $\Delta\nu^j \to 0$ in order to maximally preserve the diffusion performance. For simplicity, we define the whole part as $\nu^j$, a new control variable of the same dimensionality as $\tau^j$. Equivalently, we wish $\nu^j$ to stay close to $\frac{\tau^j - \tau^{j+1}}{\Delta\tau}$ in order to maximally preserve the performance of the diffusion model. The above model can be rewritten in terms of each state on the trajectory $\tau^j$: $\dot{\tau}_k^j = \nu_k^j$, where $\nu_k^j$ is the $k^{th}$ component of $\nu^j$. Then, we can use the CBF method to enforce the invariance of the diffusion.

We define the general form of a SafeDiffuser as the following:

**Definition 3.3** (SafeDiffuser). *A denoising diffusion procedure (1) is defined to be a SafeDiffuser if the corresponding diffusion dynamics (6) satisfy the following certificate:*

$$\frac{db(\tau_k^j)}{d\tau_k^j} \nu_k^j + h_{k,1}(j) + \alpha(b(\tau_k^j) - h_{k,2}(j)) \geq 0, \forall k \in \{0, \ldots, H\}, \forall j \in \{0, \ldots, N-1\}, \tag{7}$$

*where $h_{k,1} : \mathbb{R} \to \mathbb{R}, h_{k,2} : \mathbb{R} \to \mathbb{R}$ are two relaxation terms that ensure the diffusion procedure is not overly constrained (e.g., from the initial time T), and they are set to 0 in default. Their exact forms corresponding to different SafeDiffusers are explicitly given in the following.*

## 3.1 ROBUST-SAFE (RoS) DIFFUSER

We first present the robust-safe Diffuser, and it has the following form to show the finite-time diffusion invariance (proof is given in Appendix A.1, recall $H$ is the planning horizon, $N$ is the diffusion step):

**Theorem 3.4.** *Let the diffusion dynamics be defined as in (5) whose controllable form is defined as in (6). If the robust term $\gamma: \mathbb{R}^2 \to \mathbb{R}$ is chosen such that $\gamma(N, \varepsilon) \geq |\gamma(N, \varepsilon) - b(\tau_k^N)|e^{-\varepsilon N}, \forall k \in \{0, \ldots, H\}$ and*

$$h(\nu_k^j | \tau_k^j) \geq 0, \forall k \in \{0, \ldots, H\}, \forall j \in \{0, \ldots, N-1\}, \tag{8}$$

*where $h(\nu_k^j | \tau_k^j) = \frac{db(\tau_k^j)}{d\tau_k^j}\nu_k^j + \varepsilon(b(\tau_k^j) - \gamma(N, \varepsilon)), \varepsilon > 0$ correponds to a linear class $\mathcal{K}$ function in CBF (3), then the diffusion procedure $p_\theta(\tau^{i-1}|\tau^i), i \in \{1, \ldots, N\}$ is finite-time diffusion invariant.*

In the above, $h_{k,1}(j) = 0, h_{k,2}(j) = \gamma(N, \varepsilon)$ corresponding to Def. 3.3. We show how to construct a constraint equation 8 toward the satisfaction of safety (Def. 3.1), resulting in solving a quadratic program (QP) in diffusion process as later shown in (13).

One possible issue in the robust-safe diffusion is that if $b(\tau_k^j) \geq 0$ when $j$ is close to the diffusion step $N$, then $\tau_k^j$ can never violate the specification after diffusion step $j$. The state $\tau_k^j$ may get stuck at local traps from specifications during diffusion (see Fig. 6 of appendix), i.e., the intermediate generated sample fails to move toward high-likelihood regime via diffusion due to safety constraints. In order to address this issue, we propose a relaxed-safe diffuser and a time-varying-safe diffuser.

## 3.2 RELAXED-SAFE (ReS) DIFFUSER

In order to address the local trap problems imposed by specifications during the denoising diffusion procedure, we propose a variation of the robust-safe diffuser. We define the diffusion dynamics and their controllable form as in (5) - (6). The modified versions for CBFs are in the form:

$$h(\nu_k^j, r_k^j | \tau_k^j) := \frac{db(\tau_k^j)}{d\tau_k^j}\nu_k^j + \alpha(b(\tau_k^j)) - w_k(j)r_k^j \geq 0, k \in \{0, \ldots, H\}, j \in \{0, \ldots, N-1\}, \tag{9}$$

where $r_k^j \in \mathbb{R}$ is a relaxation variable that is to be determined (later shown in equation 13). $w_k(j) \geq 0$ is a diffusion time-varying weight on the relaxation variable such that it decreases to 0 as $j \to N_0$, $0 \leq N_0 \leq N-1$, and $w_k(j) = 0$ for all $j \leq N_0$ (see details in C.1). When $w_k(j) > 0$, the condition (9) is relaxed to reduce barrier from diffusion toward high likelihood; when it decreases to 0, the condition becomes a hard constraint. The theorem below shows the finite-time diffusion invariance (proof is given in Appendix A.2):

**Theorem 3.5.** *Let the diffusion dynamics be defined as in (5) whose controllable form is defined as in (6). If the robust term $\gamma: \mathbb{R}^2 \to \mathbb{R}$ is chosen such that $\gamma(N_0, \varepsilon) \geq |\gamma(N_0, \varepsilon) - b(\tau_k^{N_0})|e^{-\varepsilon N_0}, \forall k \in \{0, \ldots, H\}, 0 \leq N_0 \leq N-1$ and there exists a time-varying $w_k(j)$ with $w_k(j) = 0, \forall j \leq N_0$ s.t.*

$$h(\nu_k^j, r_k^j | \tau_k^j) \geq 0, \forall k \in \{0, \ldots, H\}, \forall j \in \{0, \ldots, N-1\}, \tag{10}$$

*where $h(\nu_k^j, r_k^j | \tau_k^j) = \frac{db(\tau_k^j)}{d\tau_k^j}\nu_k^j + \varepsilon(b(\tau_k^j) - \gamma(N_0, \varepsilon)) - w_k(j)r_k^j, \varepsilon > 0$ corresponds to a linear class $\mathcal{K}$ function in CBF (3), then the diffusion procedure $p_\theta(\tau^{i-1}|\tau^i), i \in \{0, \ldots, N\}$ is finite-time diffusion invariant.*

In the above, $h_{k,1}(j) = -w_k(j)r_k^j, h_{k,2}(j) = \gamma(N_0, \varepsilon)$ corresponding to Def. 3.3. Here, a relaxation variable $r$ with a time-varying weight $w$ are introduced upon Sec. 3.1 to soften safety constraints and avoid local traps with additional effort to solve for $r$ and to design $w$. This is implemented as a QP later shown in (14).

## 3.3 TIME-VARYING-SAFE (TVS) DIFFUSER

As an alternative to the relaxed-safe diffuser, we propose another safe diffuser called the time-varying-safe diffuser in this subsection. The proposed time-varying-safe diffuser can also address the local trap issues induced by specifications.

---

**Algorithm 1** Enforcing invariance in diffusion models within a diffusion step

---

**Input:** the last trajectory of diffusion $\boldsymbol{\tau}^{j+1}$ at diffusion step $j \in \{0,\dots,N\}$
**Output:** safe diffusion state $\boldsymbol{\tau}^{j*}$.
(a) Run diffusion procedure and sample as in (4) at step $j$ and get $\boldsymbol{\tau}^j$.
(b) Find diffusion dynamics as in (5) - (6).
**if** *Robust-safe diffuser* **then**
    Formulate the QP (13), solve it and get $\boldsymbol{\nu}^{j*}$.
**else if** Relaxed-safe diffuser **then**
    Define the time-varying weight $w_k(j)$ in (9), formulate the QP (14), solve it and get $\boldsymbol{\nu}^{j*}, r^{j*}$.
**else**
    Design the time-varying function $\sigma_k(j)$ in (11), formulate the QP (13), solve it and get $\boldsymbol{\nu}^{j*}$.
**end if**
(c) Update dynamics (6) with $\boldsymbol{\nu}^j = \boldsymbol{\nu}^{j*}$ and get $\boldsymbol{\tau}^{j*}$. Finally, $\boldsymbol{\tau}^j \leftarrow \boldsymbol{\tau}^{j*}$.

---

In this case, we directly modify the specification $b(\boldsymbol{\tau}_k^j) \geq 0$ by a diffusion time-varying function $\sigma_k : j \to \mathbb{R}$ (as opposed to the last two safe diffusers with a constant robust term $\gamma(N, \varepsilon)$) in the form:

$$b(\boldsymbol{\tau}_k^j) - \sigma_k(j) \geq 0, k \in \{0,\dots,H\}, j \in \{0,\dots,N\}, \tag{11}$$

where $\sigma_k(j)$ is continuously differentiable, and is defined such that $\sigma_k(N) \leq b(\boldsymbol{\tau}_k^N)$ and $\sigma_k(0) = 0$.

Finally, we have the following theorem to show the finite-time diffusion invariance:

**Theorem 3.6.** *Let the diffusion dynamics be defined as in (5) whose controllable form is defined as in (6). If there exist an extended class $\mathcal{K}$ function $\alpha$ and a time-varying function $\sigma_k(j)$ where $\sigma_k(N) \leq b(\boldsymbol{\tau}_k^N)$ and $\sigma_k(0) = 0$ s.t.*

$$h(\boldsymbol{\nu}_k^j | \boldsymbol{\tau}_k^j, \sigma_k(j)) \geq 0, \forall k \in \{0,\dots,H\}, \forall j \in \{0,\dots,N-1\}, \tag{12}$$

*where $h(\boldsymbol{\nu}_k^j | \boldsymbol{\tau}_k^j, \sigma_k(j)) = \frac{db(\boldsymbol{\tau}_k^j)}{d\boldsymbol{\tau}_k^j}\boldsymbol{\nu}_k^j - \dot{\sigma}_k(j) + \alpha(b(\boldsymbol{\tau}_k^j) - \sigma_k(j))$, then the diffusion procedure $p_\theta(\boldsymbol{\tau}^{i-1}|\boldsymbol{\tau}^i), i \in \{0,\dots,N\}$ is finite-time diffusion invariant.*

In the above, $h_{k,1}(j) = -\dot{\sigma}_k(j), h_{k,2}(j) = \sigma_k(j)$ corresponding to Def. 3.3. Here, we show an alternative to avoid local traps via a time-varying $b - \sigma_k(j)$ in contrast to Sec. 3.2 . This is implemented similar to Sec. 3.1 with additional time-varying $\sigma_k$ , later shown in (13).

**Principles of choosing the three SafeDiffusers.** We first determine if the unsafe set defined by the safety constraint is convex or not, and determine if there are any intersections between any two unsafe sets, as well as determine if the union of those intersected unsafe sets is convex or not. Then, we run the following algorithm to determine which one to choose: If all the unsafe sets are convex and the unions of intersected unsafe sets are convex (if they exist), which implies that there are no traps, then we choose the RoS diffuser; Else if we wish to have the freedom to choose nonlinear class $\mathcal{K}$ functions in designing, then we choose the TVS diffuser; Otherwise, we choose the ReS diffuser. In cases where unsafe sets are hard to determine the convexity, we may simultaneously implement the above three SafeDiffusers as they are computationally efficient (closed-form solutions are given later). Then, we can select the most desired trajectory (e.g., no local trap points) from them.

## 4 Enforcing Invariance in Diffuser

We show how we may incorporate the three proposed methods into diffusion models. In this section, we propose a minimum-deviation quadratic program (QP) approach to achieve that. We wish to enforce these conditions at every step of the diffusion as those states that are far from the specification boundaries can also be optimized accordingly, and thus, the model may generate coherent trajectories.

**Enforcing Invariance for RoS and TVS Diffusers.** During implementation, the diffusion time step length $\Delta\tau$ in (5) is chosen to be small enough, and we wish the control $\boldsymbol{\nu}^j$ to stay close to the right-hand side of (5). Thus, we can formulate the following QP-based optimization to find the optimal control for $\boldsymbol{\nu}^j$ that satisfies the condition in Thms. 3.4 or 3.6:

$$\boldsymbol{\nu}^{j*} = \arg\min_{\boldsymbol{\nu}^j} ||\boldsymbol{\nu}^j - \frac{\boldsymbol{\tau}^j - \boldsymbol{\tau}^{j+1}}{\Delta\tau}||^2, \quad \text{s.t., (8) if RoS diffuser else s.t., (12),} \tag{13}$$

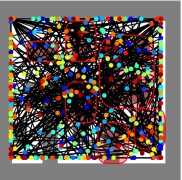 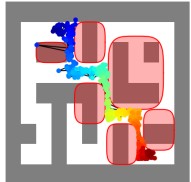 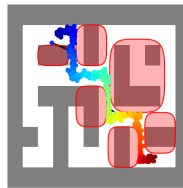 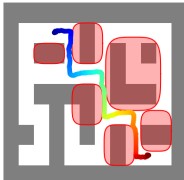

Figure 2: Maze planning (blue to red) denoising diffusion procedure with ReS diffuser in a narrow setting. Left to right: diffusion time steps 256, 5, 3, 0, respectively. Red areas denote unsafe regions. The proposed ReS diffuser can guarantee safety at the end of diffusion.

Table 2: Maze planning comparisons. Items are short for **minimum** metrics of satisfaction of simple specifications (S-SPEC) and complex specifications (C-SPEC), score of planning tasks (SCORE), computation time at each diffusion step (TIME) in seconds, and negative log likelihood (NLL), respectively. In the method column, items are short for Truncate (Trunc.), Classifier-guidance (CG), Invariant neural ODE (InvODE), relaxed-safe diffuser with last 10 step invariance (ReS-DIFFUSER-l10), respectively. The CG$-\varepsilon$ method applies (safe) gradient when the state is $\varepsilon > 0$ close to the boundary. The TRAP RATE $r$ denotes the trap rate with the number of trapped trajectory points $\geq r$.

| METHOD | S-SPEC($\uparrow$ & $\geq 0$) | C-SPEC($\uparrow$ & $\geq 0$) | SCORE ($\uparrow$) | TIME | NLL | TRAP RATE 1 ($\downarrow$) | TRAP RATE 2 ($\downarrow$) |
|---|---|---|---|---|---|---|---|
| DIFFUSER JANNER ET AL. (2022) | -0.983 | -0.894 | 1.598±0.174 | 0.006 | 4.501±0.475 | | |
| TRUNC. BROCKMAN ET AL. (2016) | $-1.192e^{-7}$ | -0.759 | 1.577±0.242 | 0.024 | 4.494±0.465 | | |
| CG DHARIWAL & NICHOL (2021) | -0.789 | -0.979 | 0.384±0.020 | 0.053 | 6.962±0.350 | | |
| CG$-\varepsilon$ DHARIWAL & NICHOL (2021) | -0.853 | -0.995 | 0.383±0.017 | 0.061 | 6.975±0.343 | | |
| InvODE XIAO ET AL. (2023B) | 14.000 | $1.657e^{-5}$ | -0.025±0.000 | 0.018 | – | | |
| RoS-DIFFUSER (OURS) | 0.010 | 0.010 | 1.519±0.330 | 0.106 | 4.584±0.646 | 100% | 100% |
| RoS-DIFFUSER-CF (OURS) | 0.010 | 0.010 | 1.536±0.306 | 0.007 | 4.481±0.298 | 100% | 100% |
| ReS-DIFFUSER (OURS) | 0.010 | 0.010 | 1.557±0.289 | 0.107 | 4.434±0.561 | 46% | 17% |
| ReS-DIFFUSER-CF (OURS) | 0.010 | 0.010 | 1.544±0.280 | 0.007 | 4.619±0.652 | 36% | 16% |
| TVS-DIFFUSER (OURS) | 0.003 | 0.003 | 1.543±0.303 | 0.107 | 4.533±0.494 | 47% | 21% |
| TVS-DIFFUSER-CF (OURS) | 0.003 | 0.003 | 1.588±0.231 | 0.007 | 4.462±0.431 | 48% | 18% |
| ReS-DIFFUSER-L10 (OURS) | 0.010 | 0.010 | 1.527±0.291 | 0.011 | 4.571±0.693 | 39% | 8% |

where $||\cdot||$ denotes the 2-norm of a vector. If we have more than one specification, we can add the corresponding conditions in Thm. 3.4 for each of them to the above QP. After we solve the above QP and get $\nu^{j*}$, we update (6) by setting $\nu^j = \nu^{j*}$ within the time step and get a new state for the diffusion procedure. Note that all of these happen at the end of each diffusion step.

**Enforcing Invariance for ReS Diffuser.** In this case, since we have relaxation variables for each of the safety specifications, we wish to minimize these relaxations in the cost function to drive all the states towards the satisfaction of specifications. In other words, we have the following QP:

$$\nu^{j*}, r^{j*} = \arg\min_{\nu^j, r^j} ||\nu^j - \frac{\tau^j - \tau^{j+1}}{\Delta\tau}||^2 + ||r^j||^2, \text{ s.t., (10),} \tag{14}$$

where $r^j$ is the concatenation of $r_k^j$ for all $k \in \{0, \ldots, H\}$. As an alternative, all the constraints above may share the same relaxation variable, i.e., the dimension of $r^j$ is only one. After we solve the QP and get $\nu^{j*}$, we update (6) by setting $\nu^j = \nu^{j*}$ within the time step and get a new state.

**Complexity/Improving efficiency.** The computational complexity of a QP is $\mathcal{O}(q^3)$, where $q$ is the dimension of the decision variable. The SafeDiffusers are more computationally expensive than existing models as they are involved with solving QPs. However, this can be addressed by: (a) Applying the proposed methods to limited diffusion steps while ensuring the satisfaction of the conditions in Thms. 3.4-3.6 are satisfied to guarantee safety; (b) Using the Batch QP solving method from the OptNet Amos & Kolter (2017); (c) Merging a number of safety constraints into a single one Lindemann & Dimarogonas (2018) or consider the most-violating two constraints at each time step, and then we can find the closed-form solution of the QP Ames et al. (2017) (See Appendix Sec. B for more details). The algorithm for enforcing invariance includes the construction of proper conditions, the solving of QP, and the update of diffusion state. We summarize the algorithm in Alg. 1.

# 5 EXPERIMENTS

We set up experiments to answer the following questions: Does our method match the theoretical potential in various tasks quantitatively and qualitatively? How does our method compare with state-of-the-art approaches in enforcing safety specifications? How does our proposed method affect the performance of diffusion under guaranteed specifications? We focus on three experiments from D4RL (Farama-foundation): maze (maze2d-large-v1), gym robots (Walker2d-v2 and Hopper-v2), and manipulation. The training data is publicly available, see Janner et al. (2022). The experiment details and metrics used are shown in Appendix. The safe diffusers generate both planning trajectory and control for the robots, and the score/reward is based on closed-loop control.

## 5.1 SAFE PLANNING IN MAZE

We focus on the case that the training data does not satisfy safety constraints to show how our methods can be generalized to new constraints. For cases where the training data satisfies safety constraints, diffusers may still violate such constraints, while our methods still work (see Fig. 8 of Appendix).

The diffuser cannot guarantee the satisfaction of any specifications. The classifier-based guidance in diffusion for safety specifications generates trajectories that largely deviate from the desired one with no safety. The proposed RoS-diffuser may introduce local trap problems (as shown in Fig. 6 of Appendix), but this can be addressed by ReS-diffuser and TVS-diffuser. The safe diffusers can all guarantee the satisfaction of specifications, even when the specifications are complex (as long as they are differentiable), as shown in Table 2. The proposed methods can also maximally preserve the performance of diffusion models, and this is demonstrated by the scores and negative log likelihood (NLL) in Table 2, as well as shown by Fig. 2 lower case. The NLL metric quantifies the similarity between different distributions, and the proposed safe diffusers can achieve similar NLL as the baseline diffuser. The computation time of safe diffusers can be significantly reduced by applying the invariance method to limited diffuser steps, as shown in the last column of Table 2 (0.011s v.s. 0.007s of diffuser). The invariant neural ODE method Xiao et al. (2023b) can guarantee safety for planning, but it does not work well in the closed-loop control (Fig. 7 of appendix), as shown by the score (-0.025) in Table 2. More ablation studies are given in Appendix C.1

## 5.2 SAFE PLANNING FOR ROBOT LOCOMOTION

Table 3: Robot safe planning comparisons with benchmarks. Abbreviations are the same as Table 2.

| EXPERIMENT | METHOD | S-SPEC($\uparrow$ & $\geq 0$) | C-SPEC($\uparrow$ & $\geq 0$) | SCORE ($\uparrow$) | TIME |
|---|---|---|---|---|---|
| WALKER2D | DIFFUSER JANNER ET AL. (2022) | -9.375 | -4.891 | 0.346±0.106 | 0.037 |
| | TRUNC. BROCKMAN ET AL. (2016) | 0.0 | $\times$ | 0.286±0.180 | 0.105 |
| | CG DHARIWAL & NICHOL (2021) | -0.575 | -0.326 | 0.208±0.140 | 0.053 |
| | ROS-DIFFUSER (OURS) | 0.000 | 0.010 | 0.312±0.165 | 0.183 |
| | ROS-DIFFUSER-CF (OURS) | 0.000 | 0.010 | 0.321±0.119 | 0.040 |
| HOPPER | DIFFUSER JANNER ET AL. (2022) | -2.180 | -1.862 | 0.455±0.038 | 0.038 |
| | TRUNC. BROCKMAN ET AL. (2016) | 0.0 | $\times$ | 0.436±0.067 | 0.046 |
| | CG DHARIWAL & NICHOL (2021) | -0.894 | -0.524 | 0.478±0.038 | 0.047 |
| | ROS-DIFFUSER (OURS) | 0.000 | 0.010 | 0.430±0.040 | 0.170 |
| | ROS-DIFFUSER-CF (OURS) | 0.000 | 0.010 | 0.464±0.028 | 0.040 |

In robot locomotion, there is no local trap problem, we only consider RoS-diffuser. Others work similarly. As expected, collisions with the roof are very likely to happen in the walker and hopper using the diffuser since there are no guarantees, as shown in Table 3. The truncation method can work for simple specifications (S-spec), but not for complex specifications (C-spec). The classifier-based guidance can improve the satisfaction of specifications but without guarantees. Collision-free is guaranteed using the RoS-diffuser, and one example of diffusion procedure is shown in Fig. 3.

## 5.3 SAFE PLANNING FOR MANIPULATION

In manipulation, specifications are joint limitations to avoid collision in joint space. In this case, the truncation method still fails to work for complex specifications (speed-dependent joint limitations).

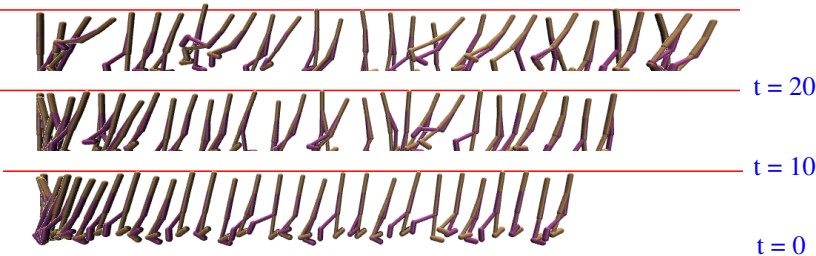

Figure 3: Walker2D planning denoising diffusion with the robust-safe diffuser (Up to down: diffusion time steps 20, 10, 0, respectively). The red line denotes the roof the walker needs to safely avoid during locomotion (safety specifications). Safety is violated at step 20 since the trajectory is initially Gaussian noise, but is eventually guaranteed (step 0). Note that the robot top could touch the roof, and this is not a collision. This can be avoided by defining a more strict safety constraint.

Table 4: Manipulation planning comparisons. Abbreviations are the same as Table 2.

| METHOD | S-SPEC($\uparrow$ & $\geq 0$) | C-SPEC($\uparrow$ & $\geq 0$) | REWARD ($\uparrow$) | TIME |
|---|---|---|---|---|
| DIFFUSER JANNER ET AL. (2022) | -0.057 | -0.065 | 0.650±0.107 | 0.038 |
| TRUNC. BROCKMAN ET AL. (2016) | $1.631e^{-8}$ | $\times$ | 0.575±0.112 | 0.069 |
| CG DHARIWAL & NICHOL (2021) | -0.050 | -0.053 | 0.800±0.328 | 0.075 |
| RoS-DIFFUSER (OURS) | 0.072 | 0.069 | 0.925±0.107 | 0.088 |
| RoS-DIFFUSER-CF (OURS) | 0.093 | 0.002 | 0.800±0.114 | 0.039 |

Our proposed RoS-diffuser can work for all specifications as long as they are differentiable. An interesting observation is that the proposed RoS-diffuser can even improve the performance (reward) of diffusion models in this case, as shown in Table 4. This may be due to the fact that the satisfaction of joint limitations can avoid collision in the joint space of the robot as Pybullet is a physics simulator. The computation time of the proposed RoS-diffuser is comparable to other methods. An illustration of the safe diffusion and manipulation procedure is shown in Fig. 4.

## 6 RELATED WORKS

**Diffusion models and planning** Diffusion models Sohl-Dickstein et al. (2015) Ho et al. (2020) are data-driven generative modeling tools, widely used in applications to image generations Dhariwal & Nichol (2021) Du et al. (2020b), in planning Hafner et al. (2019) Janner et al. (2021) Ozair et al. (2021) Janner et al. (2022), and in language Saharia et al. (2022) Liu et al. (2023a). Generative models are combined with reinforcement learning to explore dynamic models in the form of convolutional U-networks Kaiser et al. (2019), stochastic recurrent networks Ke et al. (2019), neural ODEs Du et al. (2020a), generative adversarial networks Eysenbach et al. (2022), neural radiance fields Li et al. (2022), and transformers Chen et al. (2022). Further, planning tasks are becoming increasingly important for diffusion models Lambert et al. (2021) Ozair et al. (2021) Janner et al. (2022) as they can generalize well in all kinds of robotic problems. Existing methods for improving the safety of diffusion models employ safety constraints to guide the diffusion process Yuan et al. (2022) Ajay et al. (2023) Liu et al. (2023b). However, there are no methods to equip diffusion models with safety, which is especially important for many applications. Here, we address this issue using the proposed finite-time diffusion invariance.

**Set invariance and CBFs.** An invariant set has been widely used to represent the safe behavior of dynamical systems Preindl (2016) Rakovic et al. (2005) Ames et al. (2017) Glotfelter et al. (2017) Xiao & Belta (2021). In the state of the art of control, Control Barrier Functions (CBFs) are also widely used to prove set invariance Aubin (2009), Prajna et al. (2007), Wisniewski & Sloth (2013). CBFs can be traced back to optimization problems Boyd & Vandenberghe (2004), and are Lyapunov-like functions Wieland & Allgöwer (2007). For time-varying systems, CBFs can also be adapted accordingly Lindemann & Dimarogonas (2018). Existing CBF approaches are usually applied in planning time since they are closely coupled with system dynamics. There are few studies of CBFs in other space, such as the diffusion time. Our work addresses all these limitations.

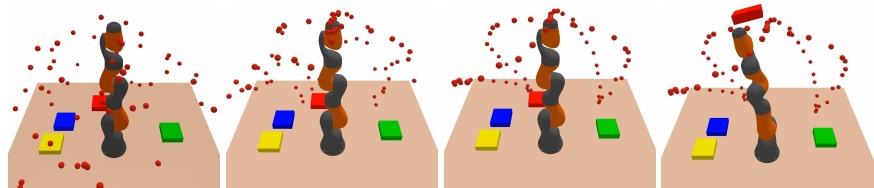

Figure 4: Manipulation planning denoising diffusion procedure with the proposed robust-safe diffuser (Left to right: diffusion time steps 1000, 100, 0, and execution time step 100, respectively). The red dots denote the planning trajectory of the end-effector.

**Guarantees in neural networks.** Differentiable optimization methods show promise for neural network controllers with guarantees Pereira et al. (2020); Amos et al. (2018); Xiao et al. (2023a). They are usually served as a layer (filter) in the neural networks. In Amos & Kolter (2017), a differentiable quadratic program (QP) layer, called OptNet, was introduced. OptNet with CBFs has been used in neural networks as a filter for safe controls Pereira et al. (2020), in which CBFs are not trainable, thus, potentially limiting the system's learning performance. In Deshmukh et al. (2019); Zhao et al. (2021); Ferlez et al. (2020), safety guaranteed neural network controllers have been learned through verification-in-the-loop training. The verification approaches cannot ensure coverage of the entire state space. More recently, CBFs have been incorporated into neural ODEs to equip them with specification guarantees Xiao et al. (2023b). However, none of these methods can be applied in diffusion models, which we address in this paper.

## 7 CONCLUSIONS, LIMITATIONS AND FUTURE WORK

We have proposed finite-time diffusion invariance for diffusion models to ensure safe planning. The proposed robust-safe diffuser can guarantee safety in general settings, but it may be subject to local trap issues. The proposed relaxed-safe and time-varying safe diffusers can address the local trap problem. Through a series of robotic planning tasks, we have demonstrated the effectiveness of our theoretical results. Our methods work better than existing approaches, while maximally preserving the model performance. Nonetheless, our method faces a few shortcomings motivating for future work.

**Limitations.** Specifically, specifications for diffusion models are expressed as differentiable constraints that may be unknown for planning tasks. Further work may explore how to learn specifications from history trajectory data Robey et al. (2020). The computation time of existing diffusion models is still too high to be applied to real robotic systems. We may explore more efficient methods in the future, such as via online replanning Zhou et al. (2023). Moreover, there may be some errors when estimating the diffusion dynamics in the current framework. This can be addressed using stochastic differential equations in diffusion models Song et al. (2020).

## ACKNOWLEDGEMENTS

The research was supported in part by Capgemini Engineering. It was also partially sponsored by the United States Air Force Research Laboratory and the United States Air Force Artificial Intelligence Accelerator and was accomplished under Cooperative Agreement Number FA8750-19-2-1000. The views and conclusions contained in this document are those of the authors and should not be interpreted as representing the official policies, either expressed or implied, of the United States Air Force or the U.S. Government. The U.S. Government is authorized to reproduce and distribute reprints for Government purposes notwithstanding any copyright notation herein. This research was also supported in part by the AI2050 program at Schmidt Futures (Grant G- 965 22-63172),

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

# A  PROOF

## A.1  PROOF OF THM. 3.4

**Proof:** Given a continuously differentiable constraint $h(\boldsymbol{x}_t) \geq 0$ ($h(\boldsymbol{x}_0) \geq 0$), by Nagumo's theorem Nagumo (1942), the necessary and sufficient condition for the satisfaction of $h(\boldsymbol{x}_t) \geq 0, \forall t \geq 0$ is

$$\dot{h}(\boldsymbol{x}_t) \geq 0, \ when \ h(\boldsymbol{x}_t) = 0,$$

If $b(\boldsymbol{\tau}_k^N) - \gamma(N, \varepsilon) \geq 0, k \in \{0, \dots, H\}$, then the condition (8) is equivalent to

$$\frac{db(\boldsymbol{\tau}_k^j)}{d\boldsymbol{\tau}_k^j} \dot{\boldsymbol{\tau}}_k^j + \varepsilon(b(\boldsymbol{\tau}_k^j) - \gamma(N, \varepsilon)) \geq 0,$$

where $\dot{\boldsymbol{\tau}}_k^j$ is the diffusion time derivative. The last equation is equivalent to

$$\frac{d(b(\boldsymbol{\tau}_k^j) - \gamma(N, \varepsilon))}{d\tau} + \varepsilon(b(\boldsymbol{\tau}_k^j) - \gamma(N, \varepsilon)) \geq 0,$$

where $\tau$ denotes the diffusion time.

Further, we have that

$$\varepsilon(b(\boldsymbol{\tau}_k^j) - \gamma(N, \varepsilon)) \rightarrow 0, \ as \ b(\boldsymbol{\tau}_k^j) \rightarrow \gamma(N, \varepsilon),$$

In other words, we have $\frac{d(b(\boldsymbol{\tau}_k^j) - \gamma(N, \varepsilon))}{d\tau} \geq 0$ when $b(\boldsymbol{\tau}_k^j) = \gamma(N, \varepsilon)$. Since $b(\boldsymbol{\tau}_k^N) \geq \gamma(N, \varepsilon), k \in \{0, \dots, H\}$, then by Nagumo's theorem, we have $b(\boldsymbol{\tau}_k^j) \geq \gamma(N, \varepsilon) > 0, \forall j \in \{0, \dots, N-1\}$. Therefore, the diffusion procedure $p_\theta(\boldsymbol{\tau}^{i-1} | \boldsymbol{\tau}^i), i \in \{1, \dots, N\}$ is finite-time diffusion invariant, and the finite time in diffusion invariance is $N$.

If, on the other hand, $b(\boldsymbol{\tau}_k^N) < \gamma(N, \varepsilon), k \in \{0, \dots, H\}$, then we can define a Lyapunov function:

$$V(\boldsymbol{\tau}_k^j) = \gamma(N, \varepsilon) - b(\boldsymbol{\tau}_k^j), k \in \{0, \dots, H\}, j \in \{0, \dots, N\}, \tag{15}$$

and $V(\boldsymbol{x}_k^N) > 0$.

Replacing $\gamma(N, \varepsilon) - b(\boldsymbol{\tau}_k^j)$ by $V(\boldsymbol{\tau}_k^j)$, the condition (8) is equivalent to (note that $\dot{\boldsymbol{\tau}}_k^j = \boldsymbol{\nu}_k^j$)

$$\frac{dV(\boldsymbol{\tau}_k^j)}{d\boldsymbol{\tau}_k^j} \dot{\boldsymbol{\tau}}_k^j + \varepsilon V(\boldsymbol{\tau}_k^j) \leq 0,$$

which is equivalent to

$$\dot{V}(\boldsymbol{\tau}_k^j) + \varepsilon V(\boldsymbol{\tau}_k^j) \leq 0,$$

Suppose we have

$$\dot{V}(\boldsymbol{\tau}_k^j) + \varepsilon V(\boldsymbol{\tau}_k^j) = 0,$$

the solution to the above equation is

$$V(\boldsymbol{\tau}_k^j) = V(\boldsymbol{\tau}_k^N) e^{-\varepsilon(N-j)},$$

Using the comparison lemma Khalil (2002), equation (8) implies that

$$V(\boldsymbol{\tau}_k^j) \leq V(\boldsymbol{\tau}_k^N) e^{-\varepsilon(N-j)}, j \in \{0, \dots, N\},$$

At diffusion step 0, i.e., $j = 0$, the last inequality becomes

$$V(\boldsymbol{\tau}_k) \leq V(\boldsymbol{\tau}_k^N) e^{-\varepsilon N}, k \in \{0, \dots, H\},$$

Substituting $V(\boldsymbol{\tau}_k^j) = \gamma(N, \varepsilon) - b(\boldsymbol{\tau}_k^j), j \in \{0, \dots, N\}$ into the last equation, we have

$$\gamma(N, \varepsilon) - b(\boldsymbol{\tau}_k) \leq (\gamma(N, \varepsilon) - b(\boldsymbol{\tau}_k^N)) e^{-\varepsilon N}, k \in \{0, \dots, H\},$$

Since $b(\tau_k^N) < \gamma(N, \varepsilon)$ in this case, the last equation can be rewritten as

$$-b(\tau_k) \leq |\gamma(N, \varepsilon) - b(\tau_k^N)|e^{-\varepsilon N} - \gamma(N, \varepsilon), k \in \{0, \ldots, H\},$$

Following the condition $\gamma(N, \varepsilon) \geq |\gamma(N, \varepsilon) - b(\tau_k^N)|e^{-\varepsilon N}$ in the theorem, we have

$$-b(\tau_k) \leq |\gamma(N, \varepsilon) - b(\tau_k^N)|e^{-\varepsilon N} - \gamma(N, \varepsilon) \leq 0, k \in \{0, \ldots, H\},$$

Therefore,

$$b(\tau_k) \geq 0, \forall k \in \{0, \ldots, H\},$$

the diffusion procedure $p_\theta(\tau^{i-1}|\tau^i), i \in \{1, \ldots, N\}$ is finite-time diffusion invariant. ∎

## A.2 PROOF OF THM. 3.5

**Proof:** Since the weight $w_k(j)$ is chosen such that $w_k(j) = 0$ for all $j \leq N_0, 0 \leq N_0 \leq N - 1$, then the condition (10) becomes a hard constraint when $j < N_0$. In other words, equation (10) becomes:

$$h(\nu_k^j|\tau_k^j) := \frac{db(\tau_k^j)}{d\tau_k^j}\nu_k^j + \alpha(b(\tau_k^j)) \geq 0, k \in \{0, \ldots, H\}, j \in \{0, \ldots, N_0\},$$

Then, the proof is similar to that of the Thm. 3.4, and we have that the diffusion procedure $p_\theta(\tau^{i-1}|\tau^i), i \in \{0, \ldots, N\}$ is finite-time diffusion invariant. ∎

## A.3 PROOF OF THM. 3.6

**Proof:** Since $\sigma_k(N) \leq b(\tau_k^N)$, we have that $s(\tau_k^j, \sigma_k(j)) := b(\tau_k^j) - \sigma_k(j) \geq 0$ when $j = N$.

The condition (12) is equivalent to

$$\frac{\partial s(\tau_k^j, \sigma_k(j))}{\partial \tau_k^j}\nu_k^j + \frac{\partial s(\tau_k^j, \sigma_k(j))}{\partial j} + \alpha(s(\tau_k^j, \sigma_k(j))) \geq 0,$$

which can be rewritten as

$$\dot{s}(\tau_k^j, \sigma_k(j)) + \alpha(s(\tau_k^j, \sigma_k(j))) \geq 0,$$

Using the Nagumo' theorem presented in the proof of Thm. 3.2, we have that

$$s(\tau_k^j, \sigma_k(j)) \geq 0, \forall j \in \{0, \ldots, N\}$$

since $s(\tau_k^N, \sigma_k(N)) \geq 0$.

As $\sigma_k(0) = 0$ and $s(\tau_k^j, \sigma_k(j)) := b(\tau_k^j) - \sigma_k(j)$, we have that $b(\tau_k^0) \geq 0, \forall k \in \{0, \ldots, H\}$. Therefore, the diffusion procedure $p_\theta(\tau^{i-1}|\tau^i), i \in \{0, \ldots, N\}$ is finite-time diffusion invariant, and the finite time in diffusion invariance is 0. ∎

## B  CLOSED-FORM SOLUTION TO SAFEDIFFUSERS

The enforcement of the proposed SafeDiffusers involve the solving of the QP (13) or (14), which could be computationally expensive for complex tasks. Here, we propose to find the closed-form solution to the QP (13) or (14) following Luenberger (1997). We take (13) with the RoS-Diffuser as an example (the closed-form solution to the TVS-Diffuser or ReS-Diffuser is similar by replacing the corresponding constraints).

Consider the following optimization corresponding to (13) for the RoS-Diffuser:

$$\boldsymbol{\nu}^{j*} = \arg\min_{\boldsymbol{\nu}^j} ||\boldsymbol{\nu}^j - \frac{\boldsymbol{\tau}^j - \boldsymbol{\tau}^{j+1}}{\Delta\tau}||^2,$$

$$\text{s.t.,} \quad \frac{db_1(\boldsymbol{\tau}^j)}{d\boldsymbol{\tau}^j}\boldsymbol{\nu}^j + \varepsilon(b_1(\boldsymbol{\tau}^j) - \gamma(N,\varepsilon)) \geq 0, \forall j \in \{0,\dots,N-1\}, \quad (16)$$

$$\frac{db_2(\boldsymbol{\tau}^j)}{d\boldsymbol{\tau}^j}\boldsymbol{\nu}^j + \varepsilon(b_2(\boldsymbol{\tau}_k^j) - \gamma(N,\varepsilon)) \geq 0, \forall j \in \{0,\dots,N-1\},$$

where $b_1, b_2$ (two vectors corresponding to the planning horizon $H$) are the two most risky safety specifications, i.e., $b_1, b_2$ have the minimum and second-minimum values (component-wise corresponding to the planning horizon $H$) among all the safety specifications at the diffusion step $j \in \{0,\dots,N-1\}$, respectively.

Define

$$g_1(\boldsymbol{\tau}^j) = [-\frac{db_1(\boldsymbol{\tau}^j)}{d\boldsymbol{\tau}^j}], \quad h_1(\boldsymbol{\tau}^j) = \varepsilon(b_1(\boldsymbol{\tau}^j) - \gamma(N,\varepsilon)),$$

$$g_2(\boldsymbol{\tau}^j) = [-\frac{db_2(\boldsymbol{\tau}^j)}{d\boldsymbol{\tau}^j}], \quad h_2(\boldsymbol{\tau}^j) = \varepsilon(b_2(\boldsymbol{\tau}^j) - \gamma(N,\varepsilon)). \quad (17)$$

The standard form of the cost in a QP is $(\nu^j)^T H(\boldsymbol{\tau}^j)\nu^j + F(\boldsymbol{\tau}^j)\nu^j$, where the matrix $H(\boldsymbol{\tau}^j)$ is an identity matrix corresponding to (16). We further define

$$[\hat{g}_1(\boldsymbol{\tau}^j), \hat{g}_2(\boldsymbol{\tau}^j)] = H(\boldsymbol{\tau}^j)^{-1}[g_1(\boldsymbol{\tau}^j), g_2(\boldsymbol{\tau}^j)],$$

$$\begin{bmatrix} \hat{h}_1(\boldsymbol{\tau}^j) \\ \hat{h}_2(\boldsymbol{\tau}^j) \end{bmatrix} = \begin{bmatrix} h_1(\boldsymbol{\tau}^j) \\ h_2(\boldsymbol{\tau}^j) \end{bmatrix} - \begin{bmatrix} g_1(\boldsymbol{\tau}^j)^T \\ g_2(\boldsymbol{\tau}^j)^T \end{bmatrix}\hat{\boldsymbol{v}}^j \quad (18)$$

where

$$\hat{\boldsymbol{v}}^j = -H(\boldsymbol{\tau}^j)^{-1}F(\boldsymbol{\tau}^j),$$

$$F(\boldsymbol{\tau}^j) = -\frac{\boldsymbol{\tau}^j - \boldsymbol{\tau}^{j+1}}{\Delta\tau}. \quad (19)$$

Then, let $\boldsymbol{w}^j := \boldsymbol{v}^j - \hat{\boldsymbol{v}}^j$ and $\langle\cdot,\cdot\rangle$ define an inner product with weight matrix $H(\boldsymbol{\tau}^j)$ so that $\langle\boldsymbol{w}^j, \boldsymbol{w}^j\rangle = (\boldsymbol{w}^j)^T H(\boldsymbol{\tau}^j)\boldsymbol{w}^j$. The optimization problem (16) is equivalent to:

$$\boldsymbol{w}^{j*} = \arg\min_{\boldsymbol{w}^j}\langle\boldsymbol{w}^j, \boldsymbol{w}^j\rangle,$$

$$\text{s.t.,} \quad \langle\hat{g}_1(\boldsymbol{\tau}^j), \boldsymbol{w}^j\rangle \leq \hat{h}_1(\boldsymbol{\tau}^j), \quad \forall j \in \{0,\dots,N-1\}, \quad (20)$$

$$\langle\hat{g}_2(\boldsymbol{\tau}^j), \boldsymbol{w}^j\rangle \leq \hat{h}_2(\boldsymbol{\tau}^j), \quad \forall j \in \{0,\dots,N-1\},$$

where the optimal solution of (16) is given by

$$\boldsymbol{\nu}^{j*} = \boldsymbol{w}^{j*} + \hat{\boldsymbol{v}}^j. \quad (21)$$

Following Luenberger (1997) [Ch. 3], the unique solution to (20) is given by

$$\boldsymbol{w}^{j*} = \lambda_1(\boldsymbol{\tau}^j)\hat{g}_1(\boldsymbol{\tau}^j) + \lambda_2(\boldsymbol{\tau}^j)\hat{g}_2(\boldsymbol{\tau}^j) \quad (22)$$

where

$$\lambda_1(\boldsymbol{\tau}^j) = \begin{cases} 0 & \text{if } G_{21}(\boldsymbol{\tau}^j)\max(\hat{h}_2(\boldsymbol{\tau}^j),0) - G_{22}(\boldsymbol{\tau}^j)\hat{h}_1(\boldsymbol{\tau}^j) < 0 \\ \frac{\max(\hat{h}_1(\boldsymbol{\tau}^j),0)}{G_{11}(\boldsymbol{\tau}^j)} & \text{if } G_{12}(\boldsymbol{\tau}^j)\max(\hat{h}_1(\boldsymbol{\tau}^j),0) - G_{11}(\boldsymbol{\tau}^j)\hat{h}_2(\boldsymbol{\tau}^j) < 0 \\ \frac{\max(G_{22}(\boldsymbol{\tau}^j)\hat{h}_1(\boldsymbol{\tau}^j) - G_{21}(\boldsymbol{\tau}^j)\hat{h}_2(\boldsymbol{\tau}^j),0)}{G_{11}(\boldsymbol{\tau}^j)G_{22}(\boldsymbol{\tau}^j) - G_{12}(\boldsymbol{\tau}^j)G_{21}(\boldsymbol{\tau}^j)} & \text{otherwise .} \end{cases}$$

$$(23)$$

$$\lambda_2(\boldsymbol{\tau}^j) = \begin{cases} \frac{\max(\hat{h}_2(\boldsymbol{\tau}^j),0)}{G_{22}(\boldsymbol{\tau}^j)} & \text{if } G_{21}(\boldsymbol{\tau}^j)\max(\hat{h}_2(\boldsymbol{\tau}^j),0) - G_{22}(\boldsymbol{\tau}^j)\hat{h}_1(\boldsymbol{\tau}^j) < 0 \\ 0 & \text{if } G_{12}(\boldsymbol{\tau}^j)\max(\hat{h}_1(\boldsymbol{\tau}^j),0) - G_{11}(\boldsymbol{\tau}^j)\hat{h}_2(\boldsymbol{\tau}^j) < 0 \\ \frac{\max(G_{11}(\boldsymbol{\tau}^j)\hat{h}_2(\boldsymbol{\tau}^j) - G_{12}(\boldsymbol{\tau}^j)\hat{h}_1(\boldsymbol{\tau}^j),0)}{G_{11}(\boldsymbol{\tau}^j)G_{22}(\boldsymbol{\tau}^j) - G_{12}(\boldsymbol{\tau}^j)G_{21}(\boldsymbol{\tau}^j)} & \text{otherwise .} \end{cases}$$

$$(24)$$

where $G(\boldsymbol{\tau}^j) = [G_{ij}(\boldsymbol{\tau}^j)] = [\langle \hat{g}_i(\boldsymbol{\tau}^j), \hat{g}_j(\boldsymbol{\tau}^j) \rangle], i, j = 1, 2$ is the Gram matrix.

## C  EXPERIMENT DETAILS

**Planning Tasks (Farama-foundation/d4rl/wiki/Tasks)**

**Maze.** In maze planning, we aim to impose trajectory constraints on the planning path of a maze. The initial positions and destinations in maze are randomly generated. The diffusion model is conditioned on the initial positions and destinations.

**Robot locomotion.** For robot locomotion (in MuJoCo), we wish the robot to avoid collisions with obstacles, such as the roof. In this case, since there is no local trap problem, we only consider robust-safe diffuser (RoS-diffuser). Others work similarly.

**Manipulation.** For manipulation (in Pybullet), the diffusion models generate joint trajectories (as controls) for the robot, which are conditioned on the locations of the objects to grasp and place. Specifications are joint limitations to avoid collision in joint space.

**Metrics and methods used in Tables**

**Metrics used in the paper Xiao et al. (2023b).** The X-SPEC, $X \in \{C, S\}$ metric is defined as:

$$\text{X-SPEC} = \min_k \{ \min_{t \in [t_0, T]} b_X(\boldsymbol{x}(t)) \}_k, k \in \{1, \ldots, N\}, \tag{25}$$

where $N$ is the number of testing runs ($N = 100$ in this case). $T$ is the final time of each run. $b_X(\boldsymbol{x}) \geq 0$ is the (complex/simple) safety constraint that is given explicitly in each experiment below.

**Classifier-based guidance** is done by applying gradients (towards safe space) to the trajectory points that drive them to the safe side of the space whenever safety constraints are violated. As the trajectory points frequently enter the unsafe sets due to the lack of set invariance property (while the CBF method does have), applying classifier-based guidance at all times could mess up the diffusion process (as shown in Fig. 2). As a result, the classifier-based guidance method would fail to identify the constraints and thus fail to satisfy them as there is no explicit constraint information involved (the CBF method does have since it evaluates the derivative of the safety constraints along the diffusion dynamics).

The **score** (results from closed-loop control) represents the normalized reward that is used in RL. The problem uses a sparse reward which has a value of 1.0 when the agent is within a 0.5 unit radius of the target. Specifically, we use the env.get_normalized_score(returns) function in dr4l to compute a normalized score for an episode, where returns are the undiscounted total sum of rewards accumulated during an episode.

**S-spec and C-spec.** Simple specifications (S-spec) and Complex specifications (C-spec) are calculated by the minimum values of the functions (e.g., $b(\tau_k)$) among all runs that define the safety constraints (e.g., $b(\tau_k) \geq 0$). They are defined by how complex the specifications are. For instance, in the maze example, the elliptical obstacle is defined as a simple obstacle as we can easily apply a truncation method to satisfy the safety constraints, while the supper-elliptical obstacle is defined as a complex obstacle as it is hard to apply the truncation method.

The **NLL** metrics for all the baselines and our methods follow the function in guided-diffusion/guided_diffusion/gaussian_diffusion.GaussianDiffusion._vb_terms_bpd from Dhariwal & Nichol (2021).

### C.1  SAFE PLANNING IN MAZE

In this experiment, we aim to impose trajectory constraints on the planning path of a maze. The training data is publicly available from Janner et al. (2022), in which initial positions and destinations in maze are randomly generated. The diffusion model is conditioned on the initial positions and destinations.

**Specifications.** The simple safety specification for the planning trajectory is defined as a super-ellipse-shape obstacle:

$$\left( \frac{x - x_0}{a} \right)^2 + \left( \frac{y - y_0}{b} \right)^2 \geq 1, \tag{26}$$

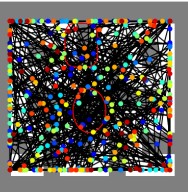 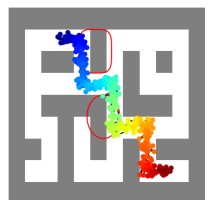 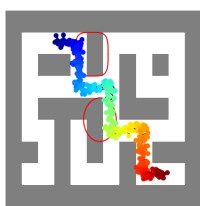 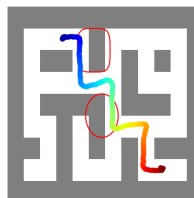

Figure 5: Maze planning (blue to red) denoising diffusion procedure with diffuser (Left to right: diffusion time steps 256, 4, 3, 0, respectively). Red ellipse and superellipse (outside) denote safe specifications. Both specifications are violated with the trajectory from diffuser.

where $(x, y) \in \mathbb{R}^2$ is the state on the planning trajectory, $(x_0, y_0) \in \mathbb{R}^2$ is the location of the obstacle. $a > 0, b > 0$. Since the state $(x, y)$ is normalized in diffusion models, we also need to normalize the above constraint accordingly. In other words, we normalize $x_0, a$ and $y_0, b$ according to the normalization of $(x, y)$ along the $x$-axis and $y-$axis, respectively.

The complex safety specification for the planning trajectory is defined as an ellipse-shape obstacle:

$$\left(\frac{x - x_0}{a}\right)^4 + \left(\frac{y - y_0}{b}\right)^4 \geq 1, \tag{27}$$

We also normalize the above constraint as in the simple case. In this case, it is non-trivial to truncate the planning trajectory to satisfy the constraint. When we have much more complex specifications, it is too hard for the truncation method to work.

**Experiment parameters.** In relaxed-safe diffuser, $w_k(j)$ is defined as $w_k(j) = 100$ when $j \geq 10$; otherwise, $w_k(j) = 0$ in the maze example. In time-varying-safe diffuser, the CBF corresponding to (26) (the CBF is similarly defined for (27)) is defined as $\left(\frac{x - x_0}{a}\right)^2 + \left(\frac{y - y_0}{b}\right)^2 \geq \sigma_k(j)$, where $\sigma_k(j) = sigmoid(j_{bias} - j)$, and $j_{bias} = 5$, in which case $\sigma_k(N)$ is near 0 at the beginning of the diffusion time $j = N$. Therefore, the unsafe set is very small such that all the trajectory points are outside the unsafe set. When $j = 0$, $\sigma_k(j)$ is close to 1, and the safety constraint is satisfied for all the trajectory points. There are many different ways to define both time-varying functions, and their definitions do not greatly affect the safety satisfaction as long as the initial (i.e., the unsafe set is very small such that all the trajectory points are initially outside the unsafe set) and terminal conditions (the safe set is the same as the safety constraint specification) are satisfied. The robust function $\gamma$ in robust-safe diffuser (the same for the relaxed-safe diffuser) is set to 0.01 according to Thm. 3.4 and the given $N$ and $\varepsilon$ ($N = 256, \varepsilon = 1$). The extended class function $\alpha$ is a linear function with slope 1 (i.e., $\varepsilon = 1$ in (8)).

**Model setup, training and testing.** The diffusion model structure is the same as the open source one (Maze2D-large-v1) provided in Janner et al. (2022). We set the planning horizon as 384, the diffusion steps as 256 for the proposed methods. The learning rate is $2e^{-4}$ with $2e^6$ training steps. The training of the model takes about 10 hours on a Nvidia RTX-3090 GPU. More parameters are provided in the attached code: "safediffuser/config/maze2d.py". The switch of different (proposed) methods in testing can be modified in "safediffuser/diffuser/models/diffusion.py" through "GaussianDiffusion.p_sample()" function.

In Fig. 5, we present a diffusion procedure using the diffuser, in which case the generated trajectory can easily violate safety constraints. Using the proposed robust-safe diffuser, the generated trajectory can guarantee safety, but some points on the trajectory may get stuck in local traps, as shown in 6. Using the proposed relaxed-safe diffuser and time-varying-safe diffuser, the local trap problem could be addressed.

**Invariant neural ODE.** The invariant neural ODE method Xiao et al. (2023b) does not work well in closed-loop control, as shown in Fig. 7.

**Ablation study 1: Maze2d-umaze-v1 (training data satisfies safety constraints).** In this case, the training data satisfies the safety constraints (modeled as maze walls). The diffuser may still violate such safety constraints due to uncertainties in inference, as shown in Fig. 8 left case. While our safe diffuser can guarantee safety (Initial positions and destinations are randomly generated).

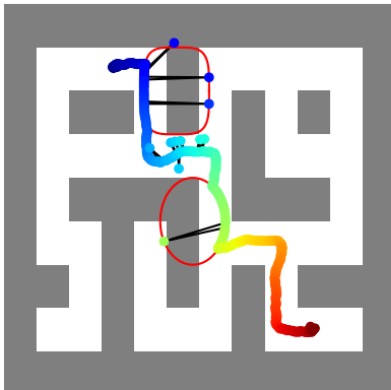

Figure 6: Maze planning (blue to red) denoising diffusion procedure with robust-safe diffuser at diffusion time step 0. Red ellipse and superellipse (outside) denote safe specifications. Although with safety guarantees, some trajectory points may get stuck in local traps.

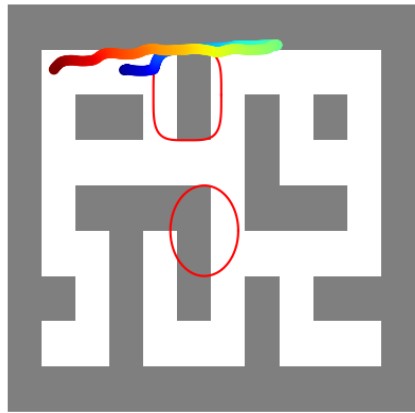

Figure 7: Maze planning (blue to red) using neural ODEs with invariance. Although with safety guarantees, the neural ODEs fail to work for such a long horizon planning problem.

**Ablation study 2: Comparison between one-step safe diffusers and multi-step safe diffusers.** The safe diffusers have to be applied for multiple diffusion steps, otherwise, the safety constraints may still be violated (as shown in Fig. 9 left case) as the proposed diffusion invariance is a dynamic process that requires several iterations to drive the trajectory to safe space.

**Ablation study 3: Safety portrait statistics.** We present in Fig. 10 the distribution of safety portrait (C-spec) among 100 runs with respect to scores. In summary, our safe diffusers can guarantee safety while maximally preserving performance.

**Ablation study 4: Minimum diffusion steps for safety.** We have done ablation studies on the number of diffusion time steps needed to reduce the computation time while still maintaining the model performance. It actually only requires at least 3 time steps (this is set to 10 in Table I to ensure robustness) as the trajectory points can quickly converge to the safe set.

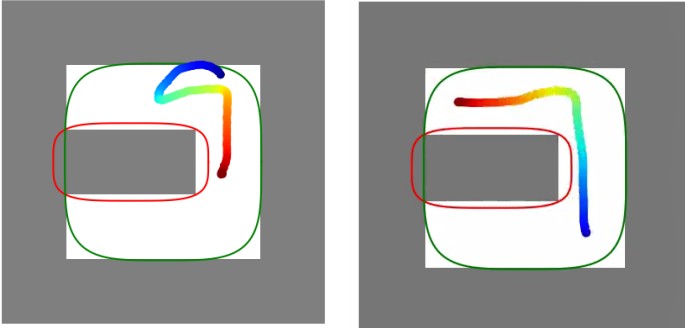

Figure 8: Maze planning (blue to red) denoising diffusion procedure using diffuser (left) and safediffuser (right) when the training data satisfies safety constraints. Red and green super-ellipses denote safe specifications for the walls. Diffusers may still violate safety constraints.

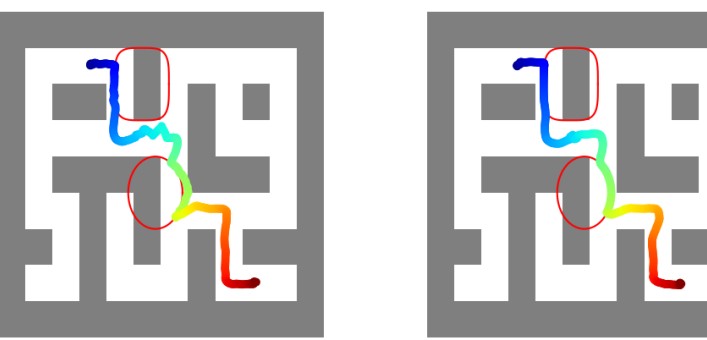

Figure 9: Maze planning (blue to red) denoising diffusion procedure using safediffuser for the last step (left) and the last 10 steps (right). The generated trajectory is more smooth with safety guarantees when running for the last 10 steps.

### C.2 SAFE PLANNING FOR ROBOT LOCOMOTION

For robot locomotion (in MuJoCo), we wish the robot to avoid collisions with obstacles, such as the roof. In this case, since there is no local trap problem, we only consider robust-safe diffuser (RoS-diffuser). Others work similarly. The training data set is publicly available from Janner et al. (2022).

**Specifications.** The simple safety specification for both the Walker2D and Hopper is collision avoidance with the roof. In other words, the height of the robot head $z \in \mathbb{R}$ should satisfy the following constraint:

$$z \leq h_r, \tag{28}$$

where $h_r > 0$ is the height of the roof. We also need to normalize $h_r$ according to the normalization of the state $z$ in the diffusion model.

The complex safety specification for both the Walker2D and Hopper is a speed-dependent collision avoidance constraint:

$$z + \varphi v_z \leq h_r, \tag{29}$$

where $\varphi > 0$, $v_z \in \mathbb{R}$ is the speed of the robot head along the $z$-axis. The speed-dependent safety constraint is more robust for the robot to avoid collision with the roof since when the robot jumps faster, we need to ensure a larger safe distance with respect to the roof in order to account for all kinds of uncertainties or perturbations. In this case, the simple truncation method is hard to work since it is not clear how to truncate both $z$ and $v_z$ at the same time.

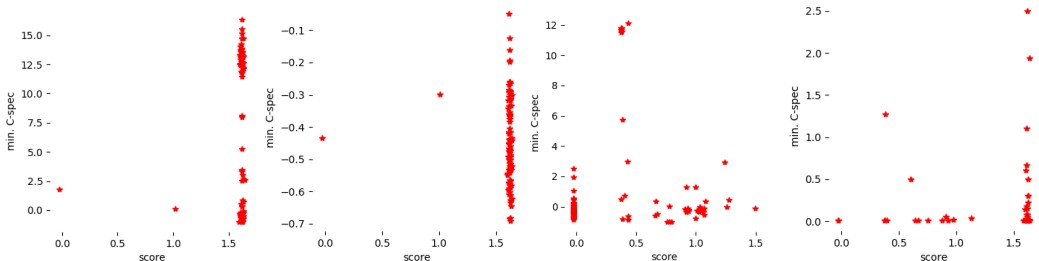

Figure 10: Specification satisfaction metrics (C-spec) v.s. scores in maze planning using diffusion models (from left to right: diffuser, truncation, classifier guidance, safediffuser). Safediffusers can guarantee the satisfaction of specifications while making the scores consistently close to 1.6 among all the runs (mean value: 1.527).

**Model setup, training and testing.** The diffusion model structures are the same as the open source ones (Walker2D-Medium-Expert-v2 and Hopper-Medium-Expert-v2) provided in Janner et al. (2022). We set the planning horizon as 600, the diffusion steps as 20. The learning rate is $2e^{-4}$ with $2e^{6}$ training steps. The training of the model takes about 16 hours on a Nvidia RTX-3090 GPU. More parameters are provided in the attached code: "safediffuser/config/locomotion.py". The switch of different methods in testing can be modified in "safediffuser/diffuser/models/diffusion.py" through "GaussianDiffusion.p_sample()" function.

## C.3  SAFE PLANNING FOR MANIPULATION

For manipulation (in Pybullet), the diffusion models generate joint trajectories (as controls) for the robot, which are conditioned on the locations of the objects to grasp and place. The training data set is publicly available from Janner et al. (2022). Specifications are joint limitations to avoid collision in joint space.

**Specifications.** The simple safety specification for the robot is in the joint space, and we are trying to limit the joint angles of the robot within allowed ranges:

$$x_{min} \le x \le x_{max}, \tag{30}$$

where $x \in \mathbb{R}^7$ is the state of 7 joint angles, $x_{min} \in \mathbb{R}^7$ and $x_{max} \in \mathbb{R}^7$ denotes the minimum and maximum joint limits. We need to normalize the limits according to how the state $x$ is normalized in the diffusion model.

The complex safety specifications are speed-dependent joint constraints:

$$x_{min} \le x + \varphi v \le x_{max}, \tag{31}$$

where $\varphi > 0$, $v \in \mathbb{R}^7$ is the joint speed corresponding to the joint angle $x$. In this example, since the diffusion model does not directly predict $v$, we evaluate $v$ using $x(k)$ and $x(k+1)$ along the planning horizon. The joints limits are also normalized as in the simple specification case.

**Model setup, training and testing.** The diffusion model structure is the same as the open source one provided in Janner et al. (2022), and we use their pre-trained models to evaluate our methods when comparing with other approaches.

