# OpenReview forum: "SafeDiffuser: Safe Planning with Diffusion Probabilistic Models"
_ICLR.cc/2025/Conference — ICLR 2025 Poster_

### Official Review · Reviewer_gj8X · 2024-11-01

**Soundness:** 2
**Presentation:** 2
**Contribution:** 2
**Rating:** 5
**Confidence:** 3

**Summary:**

The paper introduces SafeDiffuser, a method to add safety guarantees to diffusion models when used for planning tasks. The key innovation is incorporating control barrier functions (CBFs) into the diffusion process to ensure generated trajectories satisfy safety constraints.

**Strengths:**

The approach maintains the model performance while adding safety guarantees.

**Weaknesses:**

1. The definition of diffusion invariance seems too strict. Essentially we only care about the safety at t=0, and trajectories satisfying t=0 safety don’t necessarily need to satisfy t=1 safety. However the method actively encourages the intermediate trajectories to be safe. This doesn’t mean the method is wrong, but I’m hoping to see a softer way to guarantee the safety. See Q2.
2. Along with the above weakness, the design choice seems too complex. There’re $\epsilon$, $w$, $r$, $\gamma$, $\sigma$, all of them seem hyper-parameters that need careful tuning.
3. CBF itself is meant to design for dynamics that are (1) time-invariant; and (2) deterministic. The diffusion process doesn’t belong to any of these two parts. For (1), the alpha bar changes over time steps. For (2), most of the diffusion sampler constantly adds noise to the dynamics (unless the approach is specifically using DDIM). Applying CBF-based technique to the diffusion seems counter-intuitive.
4. Scaling up seems challenging. I’m hoping to see a highly nonlinear dynamics like drone/quadcopter to see whether the method could scale up. Another challenge for scaling up is from the selection criteria. Page 6 discusses the principle to select different safediffusers. As mentioned, it requires the unsafe set to be known to determine the convexity. This seems challenging for high-dimensional control tasks.

**Questions:**

1. Regarding to the design choice, the TVS diffuser seems to be the most general safediffuser. Is it possible to further simplify the forward invariance equation (theorem 3.4) by fully discarding the reliance on b, and directly using sigma? For example, imagine sigma at t=0 converges to exactly b. This sigma can be used in analogy to the time-varying CBF and should impose a more relaxed restriction to the intermediate samples. I guess it would need clever design of sigma (or even learning-based), but do you think it is worth the effort?
2. Another question is the motivation of using CBF. From first principle, all we care about here is the safety at the final timestep, ie, t=0. It sounds more similar to a goal-reaching/liveness problem, where you don’t need to enforce every intermediate sample satisfy the safety. Compared to control Lyapunov function or reachability-based method, why is CBF used for the problem? Applying the other two concepts seem more natural to me.
3. In control, CBF can usually be compared to potential-field based method to show their effective in terms of avoiding obstacle in a longer time horizon rather than reactively avoiding. Similarly, is it possible to include [1] as a baseline to show the approach’s effectiveness?

[1] Luo, Y., Sun, C., Tenenbaum, J. B., & Du, Y. (2024). Potential based diffusion motion planning. arXiv preprint arXiv:2407.06169.

---

> ### Author Response · Authors · 2024-11-18
> **Response to review comments part I**
>
> We thank the reviewer for all the constructive comments. We address all the reviewer’s concerns in the response below.
>
> **Weakness 1:** The definition of diffusion invariance seems too strict. Essentially we only care about the safety at t=0, and trajectories satisfying t=0 safety don’t necessarily need to satisfy t=1 safety. However the method actively encourages the intermediate trajectories to be safe. This doesn’t mean the method is wrong, but I’m hoping to see a softer way to guarantee the safety. See Q2.
>
> **Response:** If we apply the proposed methods to all diffusion steps (as in the RoS-diffuser), this is surely too strict. However, we have extensively studied how to address such an issue by proposing the ReS-diffuser and TVS diffuser. For instance, in TVS-diffuser, the size of unsafe sets is almost 0 at the initial diffusion time step T, which means that all the trajectories are within safe sets (the initial condition of diffusion invariance is satisfied). As we have defined a time-varying component in TVS-diffuser (shown in Thm. 3.4), the size of unsafe sets will grow with the diffusion time. At the final diffusion time step 0, the size of unsafe sets is the same as the original unsafe sets, then the safety is guaranteed. Therefore, the definition of diffusion invariance in TVS-diffuser is not strict due to the existence of the time component.  Alternatively, we may address this issue by applying safediffusers to limited diffusion steps, and we can still guarantee the safety while not being overly conservative, as shown in the ReS-diffuser-l10 of Table 2.
>
> **Weakness 2:** Along with the above weakness, the design choice seems too complex. All of them seem hyper-parameters that need careful tuning.
>
> **Response:**  The robust term (and all the parameters) is actually used to define a more strict safety specification compared to the original safety constraint. This is to overcome the asymptotic convergence properties of the CBF method. In practice, we may actually set the robust term to 0, and the resulting safety metrics (S-spec and C-spec in tables 2-4) are about $-1e^{-7}$ (they should be $\geq 0$), which shows a slightly violation of safety. Therefore, if we set the robust term to a small positive number (e.g., 0.01), we no longer need to tune these parameters. The design choice of parameters is to show formal guarantees from a theoretical perspective (providing a tight lower bound for the robust term, i.e., introducing less restrictive safety guarantees).
>
>
> **Weakness 3:** CBF itself is meant to design for dynamics that are (1) time-invariant; and (2) deterministic. The diffusion process doesn’t belong to any of these two parts. For (1), the alpha bar changes over time steps. For (2), most of the diffusion sampler constantly adds noise to the dynamics (unless the approach is specifically using DDIM). Applying CBF-based technique to the diffusion seems counter-intuitive.
>
> **Response:** We wish to clarify the misunderstanding of CBFs and our approaches here. CBFs have already been designed for dynamics/constraints that are time-varying (e.g, [1]) and stochastic (e.g., [2]).  Therefore, CBFs can be applied to the diffusion process. In fact, our proposed Time-Varying Safe (TVS)-diffuser belongs to the time-varying category. In our approaches, since we wish to strictly ensure the safety of diffusion models, we take the diffusion rollout result (during inference, a denoising process without adding noise) at each diffusion step as a reference in our QP solution, and optimally revise the diffusion process. In such a way, we can ensure the safety of diffusion models. This has been further verified on all the experiments (with code attached in the submission). Our approaches are mainly used during the inference of diffusion, and thus are not subject to the problem in (2).
>
> [1] Lindemann, Lars, et,al. "Control barrier functions for signal temporal logic tasks." IEEE control systems letters 3.1 (2018): 96-101.
>
> [2] Clark, Andrew. "Control barrier functions for stochastic systems." Automatica 130 (2021): 109688.

---

> ### Author Response · Authors · 2024-11-18
> **Response to review comments part II**
>
> **Weakness 4:** Scaling up seems challenging. I’m hoping to see a highly nonlinear dynamics like drone/quadcopter to see whether the method could scale up. Another challenge for scaling up is from the selection criteria. Page 6 discusses the principle to select different safediffusers. As mentioned, it requires the unsafe set to be known to determine the convexity. This seems challenging for high-dimensional control tasks.
>
> **Response:**  Scaling up is indeed a challenge in our previous results as we need to solve QPs in the previous methods. For highly nonlinear dynamics, the computation issue is the main concern. However,  We have fully addressed the computational issue by finding their closed-form solutions (referring to CH3 of the book ``Optimization by Vector Space Methods’’).  The closed-form solution is given in Appendix section B of the revised manuscript. We have rerun all the experiments using the closed-form solution of safediffusers, which shows that the computation time is almost the same as the original diffuser (given in the following three tables).  The dynamics of manipulators and walkers are comparable to drones, and the dimension of the state space in the walker2d (usually 17) is in fact higher than that of a drone (usually 12). This scaling capability totally depends on the diffusion models, and this has been extensively tested in the literature.
>
> The unsafe set needs to be known.  However, we may learn the safety constraints using the methods such as  in [1], and combine it with the diffuser. We can then further check the convexity of the unsafe sets from learned safety constraints. This is an ongoing research direction.  Please note that the safety is usually not defined on the control, but on the state. The CBFs can map state constraints to control constraints. For high-dimensional control tasks, this won’t be an issue using the new proposed closed-form solution in terms of computational complexity.
>
> [1] Robey, Alexander, et al. "Learning control barrier functions from expert demonstrations."  IEEE-CDC 2020.
>
> Maze planning (CF is short for closed-form solution):
> |  method |  S-spec($\geq0$)  | C-spec($\geq0$) | Score ($\uparrow$)  | Time ($\downarrow$) | NLL|  Trap rate 1| Trap rate 2|
> |---|---|---|---|---|---|---|---|
> | Diffuser |-0.983 | -0.894 |$1.598{\small\pm 0.174}$ | ${\color{red}0.006}$| 4.501| | |
> | Trunc |$-1.192e^{-7}$ | -0.759 |$1.577{\small\pm 0.242}$ | 0.024| 4.494| | |
> | CG |-0.789 | -0.979 |$0.384{\small\pm 0.020}$ | 0.053| 6.962| | |
> | CG-$\epsilon$ |-0.853 | -0.995 |$0.383{\small\pm 0.017}$ | 0.061| 6.975| | |
> | InvODE|14.000 | $1.657e^{-5}$ |$-0.025{\small\pm 0.000}$ | 0.018| - | | |
> | RoS-diffuser (ours)|0.010 | 0.010 |$1.519{\small\pm 0.330}$ | 0.106| 4.584|$100\%$ | $100\%$|
> | RoS-diffuser-CF (ours)|0.010 | 0.010 |$1.536{\small\pm 0.306}$ | ${\color{purple}0.007}$| 4.481|$100\%$ | $100\%$|
> | ReS-diffuser (ours)|0.010 | 0.010 |$1.557{\small\pm 0.289}$ | 0.107| 4.434|$46\%$ | $17\%$|
> | ReS-diffuser-CF (ours)|0.010 | 0.010 |$1.544{\small\pm 0.280}$ | ${\color{purple}0.007}$| 4.619|$36\%$ | $16\%$|
> | TVS-diffuser (ours)|0.003 | 0.003 |$1.543{\small\pm 0.303}$ | 0.107| 4.533|$47\%$ | $21\%$|
> | TVS-diffuser-CF (ours)|0.003 | 0.003 |$1.588{\small\pm 0.231}$ | ${\color{purple}0.007}$| 4.462|$48\%$ | $18\%$|
> | ReS-diffuser-l10 (ours)|0.010 | 0.010 |$1.527{\small\pm 0.291}$ | 0.011| 4.571|$39\%$ | $8\%$|
>
>
> Locomotion (CF is short for closed-form solution):
> |Experiment|  method |  S-spec($\geq0$)  | C-spec($\geq0$) | Score ($\uparrow$)  | Time($\downarrow$) |
> |---|---|---|---|---|---|
> | | Diffuser |-9.375 | -4.891 |$0.346{\small\pm 0.106}$ | ${\color{red}0.037}$|
> | | Trunc |0.0 | $\times$ |$0.286{\small\pm 0.180}$ | 0.105|
> |Walker2D| CG |-0.575 | -0.326 |$0.208{\small\pm 0.140}$ | 0.053|
> | | RoS-diffuser (ours)|0.000 | 0.010 |$0.312{\small\pm 0.165}$ | 0.183|
> | | RoS-diffuser-CF (ours)|0.000 | 0.010 |$0.321{\small\pm 0.119}$ | ${\color{purple}0.040}$|
> |---|---|---|---|---|---|
> | | Diffuser |-2.180 | -1.862 |$0.455{\small\pm 0.038}$ | ${\color{red}0.038}$|
> | | Trunc |0.0 | $\times$ |$0.436{\small\pm 0.067}$ | 0.046|
> |Hopper| CG |-0.894 | -0.524 |$0.478{\small\pm 0.038}$ | 0.047|
> | | RoS-diffuser (ours)|0.000 | 0.010 |$0.430{\small\pm 0.040}$ | 0.170|
> | | RoS-diffuser-CF (ours)|0.000 | 0.010 |$0.464{\small\pm 0.028}$ | ${\color{purple}0.040}$|
>
> Manipulation (CF is short for closed-form solution):
> |  method |  S-spec($\geq0$)  | C-spec($\geq0$) | Score ($\uparrow$)  | Time($\downarrow$) |
> |---|---|---|---|---|
> | Diffuser |-0.057 | -0.065 |$0.650{\small\pm 0.107}$ | ${\color{red}0.038}$|
> | Trunc |$1.631e^{-8}$ | $\times$ |$0.575{\small\pm 0.112}$ | 0.069|
> | CG |-0.050 | -0.053 |$0.800{\small\pm 0.328}$ | 0.075|
> | RoS-diffuser (ours)|0.072 | 0.069 |$0.925{\small\pm 0.107}$ | 0.088|
> | RoS-diffuser-CF (ours)|0.093 | 0.002 |$0.800{\small\pm 0.114}$ | ${\color{purple}0.039}$|

---

> ### Author Response · Authors · 2024-11-18
> **Response to review comments part III**
>
> **Question 1:** Regarding to the design choice, the TVS diffuser seems to be the most general safediffuser. Is it possible to further simplify the forward invariance equation (theorem 3.4) by fully discarding the reliance on b, and directly using sigma? For example, imagine sigma at t=0 converges to exactly b. This sigma can be used in analogy to the time-varying CBF and should impose a more relaxed restriction to the intermediate samples. I guess it would need clever design of sigma (or even learning-based), but do you think it is worth the effort?
>
> **Response:** Please note that b is the original safety specification, if we can define a sigma that converges to exactly b at t = 0, then we may discard the reliance on b. However, since the TVS-diffuser also has some time-varying function, it can also relax the restriction to the intermediate samples. Learning such a sigma would be interesting, and it is similar to learning a CBF [1]. It is definitely worth doing it as it can relax the assumption that b is known (see response to Weakness 4).
>
> [1] Robey, Alexander, et al. "Learning control barrier functions from expert demonstrations." 2020 59th IEEE Conference on Decision and Control (CDC). IEEE, 2020.
>
> **Question 2:** Another question is the motivation of using CBF. From first principle, all we care about here is the safety at the final timestep, ie, t=0. It sounds more similar to a goal-reaching/liveness problem, where you don’t need to enforce every intermediate sample satisfy the safety. Compared to control Lyapunov function or reachability-based method, why is CBF used for the problem? Applying the other two concepts seem more natural to me.
>
>
> **Response:** A control Lyapunov function (CLF) is similar to a CBF (it is arguably true in the literature that a CLF is a special case of a CBF, i.e., shrinking an unsafe set in a CBF to a point would generate a CLF). However, CLF is mostly used for system stability, while CBFs are used for safety (the motivation of using CBFs instead of CLFs). The reachability-based methods are known to be computationally expensive, and thus, it is challenging to use them in diffusion models, while the CBF methods are highly efficient (the main motivation of using CBFs).  Our approaches do not need to be enforced at every intermediate step, see response to Weakness 1.
>
> **Question 3:** In control, CBF can usually be compared to potential-field based method to show their effective in terms of avoiding obstacle in a longer time horizon rather than reactively avoiding. Similarly, is it possible to include [1] as a baseline to show the approach’s effectiveness?
>
> **Response:**  This is definitely a good baseline. We have added new experiments to compare the potential-field based method and our methods (see the table below). In summary, potential-field based methods have no strict safety guarantees (see Fig. 5 a in [1] showing there are still collisions with potential-based methods), while our methods do. The main disadvantage of CBFs (compared to potential-field methods) lies in its myopic property, as mentioned by the reviewer. However, this disadvantage may be addressed when CBFs are combined with diffusion models as those models are powerful to generate long time horizon (and potentially optimal) trajectories, as shown by the scores or reward metrics in Tables 2-4 of the paper.
>
> Maze planning:
> |  method |  S-spec($\geq0$)  | C-spec($\geq0$) | Score ($\uparrow$)  | Time($\downarrow$) | NLL|
> |---|---|---|---|---|---|
> | Potential-based diffuser |-0.798 | -0.908 |$0.916{\small\pm 0.556}$ | 0.015| 5.289|
> | RoS-diffuser-CF (ours)|0.010 | 0.010 |$1.536{\small\pm 0.306}$ | $0.007$| 4.481|
> | ReS-diffuser-CF (ours)|0.010 | 0.010 |$1.544{\small\pm 0.280}$ | $0.007$| 4.619|
> | TVS-diffuser-CF (ours)|0.003 | 0.003 |$1.588{\small\pm 0.231}$ | $0.007$| 4.462|

---

> ### Author Response · Authors · 2024-11-25
> **Encourage in discussion**
>
> Dear reviewer gj8X,
>
> As the deadline of the rebuttal is approaching, could you please let us know if you have any further concerns?
>
> Looking forward to your reply,
>
> Best regards,
>
> Authors.

---

> ### Author Response · Authors · 2024-11-26
> **Could you please let us know your feedback**
>
> Dear Reviewer gj8X,
>
> Thank you once again for your time and effort in reviewing our paper. We have worked very hard to address all your concerns, including new experiments showing the advantages of our methods.  Could you please kindly let us know if you still have any other questions or concerns?
>
> Much appreciated,
>
> Best regards,
>
> Authors.

---

> > ### Comment · Reviewer_gj8X · 2024-11-27
> > **Score Raised to 5**
> >
> > Thanks for addressing most of my concerns about the time-varying thing and rechability-based methods. Some of the rest concerns:
> >
> > 1. Scaling up
> >
> > The concern of scaling up is not from computation, but rather how to design a relatively well CBF for highly nonlinear dynamics that need agile controls. Since the work uses hand-crafted CBF, the concern is how this would apply if the dynamics go highly nonlinear. For the robot arm and maze environment, I think the dynamics are still relatively linear, and a manual design of CBF is relatively not challenging. The dimension is not really a problem here, but the nonlinearity is.
> >
> > 2. General stochastic sampler
> >
> > The author mentions `we take the diffusion rollout result (during inference, a denoising process without adding noise) `. I agree some reverse samplers are deterministic, like DDIM, DPM, or other Probability flow based solver. However, in general, the reverse process can also be stochastic, like DDPM. I'm wondering how the deterministic CBF would apply to those stochastic samplers like DDPM.
> >
> > 3. Lipschitz continuity
> >
> > I would disagree the forward process $x_t=\sqrt{\bar\alpha}x_0+\sqrt{1-\bar\alpha}\epsilon$ is Lipschitz continuous dynamics, because $\epsilon$ is from a normal distribution, which should be unbounded theoretically. This means the Lipschitz constant (if there's one) would be greater than any specific finite value.
> >
> > The reverse process, if it is stochastic like DDPM, would also not be Lipschitz continuous. There are Gaussian noises injected during the process, which are unbounded.
> >
> > Again, I believe this paper points in a valuable direction. But, I still think it is worth more investigation. There are 3 variants, and it doesn't seem very easy to decide which one to use and how to design a good CBF without knowing the safe set. I wish to see a simplified and unified version that takes the essence of all these 3 variants.

---

> > > ### Author Response · Authors · 2024-11-27
> > > **Thank you for raising the score**
> > >
> > > We really appreciate the reviewer for raising the score. In the below, we address all the remaining concerns/comments.
> > >
> > > **1.Scaling up: Drone v.s. manipulator (and walker)**
> > >
> > > The dynamics of drones and manipulators (and walkers) are all in 3D space, therefore, we would not agree that the dynamics of a manipulator is relatively linear to the ones of the drone. They are comparable in terms of the complexity or nonlinearity. Nonetheless, the CBF method can easily deal with highly nonlinear systems and agile control problems, see [1]. However, we totally agree with the reviewer that our work would be more impactful and stronger if we include more benchmarks and dataset (including drones), as also mentioned by reviewer hg4A. We would work on such benchmarks and dataset in the follow-up work.
> > >
> > > [1] Singletary, Andrew, et al. "Onboard safety guarantees for racing drones: High-speed geofencing with control barrier functions." IEEE Robotics and Automation Letters 7.2 (2022): 2897-2904.
> > >
> > > **2. General stochastic sampler**
> > >
> > > Our methods also work for stochastic sampler (such as DDPM) as well since  (1) our proposed SafeDiffusers act as another layer of safety filter for the denoising diffusion procedure. In other words, no matter whether the sampler is deterministic or stochastic, we just need to optimally modify the generated trajectory (deterministic or stochastic rollout) at each diffusion step; (2) our proposed SafeDiffusers work similarly if we indeed want CBFs to be fully integrated with the diffusion model (e.g., changing inner components of stochastic differential equations) instead of acting as a safety filter.  This can be done using stochastic CBFs [2].
> > >
> > > [2] Clark, Andrew. "Control barrier functions for stochastic systems." Automatica 130 (2021): 109688.
> > >
> > > **3. Lipschitz continuity**
> > >
> > > Theoretically, continuous time diffusion models (such as those based on stochastic differential equations) are not Lipschitz due to the existence of the random variables with infinite support (e.g., a normal distribution).  Practically, continuous time diffusion models are **indeed Lipschitz** as the sampling has to been done within bounded distribution during implementation. The Lipschitz constant may be large near the diffusion time 0, but they are still Lipschitz. This has been fully studied in [3] mentioning  at the end of page 4 that ``Specifically, we consider the interval near the zero point suffering from large Lipschitz constants''. Therefore, the diffusion models are mostly considered to be Lipschitz, and this is not a concern. Even in the control community, Lipschitz continuity is not a concern for stochastic systems in order to use CBFs to ensure safety [2]. We changed the clarification in the revision.
> > >
> > > [3] Yang, Zhantao, et al. "Lipschitz singularities in diffusion models." The Twelfth International Conference on Learning Representations. 2023.
> > >
> > > **4. Unified SafeDiffuser**
> > >
> > > We guess this is the main concern of the reviewer. We totally agree with the reviewer that a unified SafeDiffuser is very important for practitioners. In order to fully address this, we **added a new definition (Def. 3.3 in the revised PDF) for the SafeDiffuers that unify all the existing three SafeDiffusers**. This unified SafeDiffuser definition is based on the most general form TVS-Diffuser. We also added in the revision how each SafeDiffuser corresponds to the unified form of SafeDiffuser in Def. 3.3 (In other words, the exact formulations of the $h_{k,1}, h_{k,2}$ functions in Def. 3.3 that ensure the diffusion procedure is not overly constrained, e.g., from the initial denoising diffusion time T).
> > >
> > > We would also agree with the reviewer that it might not be easy to decide which safediffuser to choose for some safe sets. In cases where unsafe sets are hard to determine the convexity, we may simultaneously implement the above three SafeDiffusers based on the unified SafeDiffuser in Def. 3.3 as they are computationally efficient (closed-form solutions are given, the reviewer also agreed that the computation is not a concern). Then, we can select the most desired/best trajectory (e.g., no local trap points) from them. We added this clarification at the end of Sec. 3 of the revision.
> > >
> > > ----------------------------------------------------------------------------------------------------------------
> > > Please do let us know if you have any further comments/concerns. We are committed to address all the potential problems and improve our paper following your instruction. Thank you once again for your timely response.

---

> > > ### Author Response · Authors · 2024-12-02
> > > **Please let us know your further feedback**
> > >
> > > Dear reviewer gj8X,
> > >
> > > Thank you very much again for taking the time in reviewing our paper and in the rebuttal. We have worked very hard to address your concerns, especially your main concern regarding a unified and simple definition of the SafeDiffuser. We have added a formal definition (Def. 3.3 in the revised pdf) for the proposed SafeDiffuser, and have made connections to the three SafeDiffusers that address the conservativeness of the method.
> > >
> > > Could you please kindly let us know if you have any other concerns/questions?
> > >
> > > Thank you and best regards,
> > >
> > > Authors

---

### Official Review · Reviewer_vnZo · 2024-11-04

**Soundness:** 3
**Presentation:** 3
**Contribution:** 3
**Rating:** 8
**Confidence:** 4

**Summary:**

This paper presents control barrier function-based methods to enforce safety constraints on diffusion models. The main idea is to modify the diffusion denoizing step to satisfy safety constraints within a finite number of steps (finite-time diffusion invariance). This is done by formulating a quadratic program to compute a perturbation that steers the generated trajectory towards the safe set in every denoizing step. Evaluations in simulated maze and robotics domains show that the proposed method is superior to baselines in satisfying safety constraints at the cost of longer planning time.

**Strengths:**

- Diffusion models are an important and effective class of conditional generative models. However, existing methods for guidance do no provide safety guarantees. Hence, the propose algorithm is an important contribution.

- Applying CBF-based technique to diffusion models is novel to my knowledge. In particular, modifying the denoizing steps seem to be more effective than generating safe trajectories through classifier-free guidance.

- Proposed extension to handle local traps are effective.

- Experimental results in simulation demonstrate the effectiveness and robustness of the algorithm.

**Weaknesses:**

- Safety specifications have to be differentiable.

- SafeDiffuser is significantly slower (by an order of magnitude) than vanilla diffusion.

**Questions:**

- The forward invariance in control theory (equation 2) assumes that the control system is governed by Lipschitz functions. Is any such requirement not required for extending the definition to diffusion neural networks?

- Diffusion dynamics is estimated online. How does error in this estimation affect the guarantees and performance in practice?

- In theorem 3.2 what is the intuitive meaning of the robust term?

- If would be interesting to continually retrain the diffusion models on the generated safe data. Can this improve performance or speed up safediffuser?

---

> ### Author Response · Authors · 2024-11-18
> **Response to review comments part I**
>
> We appreciate the reviewer for all the positive and constructive comments.  All the concerns are addressed carefully in the responses below.
>
> **Weakness 1:** Safety specifications have to be differentiable.
>
> **Response:** We agree that this is one of the concerns. However, we may always use differentiable constraints to slightly over-approximate non-differentiable safety specifications (e.g., using a super-ellipse to cover an unsafe set with square shape).
>
> **Weakness 2:** SafeDiffuser is significantly slower (by an order of magnitude) than vanilla diffusion.
>
> **Response:** We have fully addressed the computational issue by finding their closed-form solutions (referring to CH3 of the book ``Optimization by Vector Space Methods’’).  The closed-form solution is given in Appendix section B of the revised manuscript. We have rerun all the experiments using the closed-form solution of safediffusers, which shows that the computation time is almost the same as the original diffuser (given in the following three tables).
>
> Maze planning (CF is short for closed-form solution):
> |  method |  S-spec($\geq0$)  | C-spec($\geq0$) | Score ($\uparrow$)  | Time ($\downarrow$) | NLL|  Trap rate 1| Trap rate 2|
> |---|---|---|---|---|---|---|---|
> | Diffuser |-0.983 | -0.894 |$1.598{\small\pm 0.174}$ | ${\color{red}0.006}$| 4.501| | |
> | Trunc |$-1.192e^{-7}$ | -0.759 |$1.577{\small\pm 0.242}$ | 0.024| 4.494| | |
> | CG |-0.789 | -0.979 |$0.384{\small\pm 0.020}$ | 0.053| 6.962| | |
> | CG-$\epsilon$ |-0.853 | -0.995 |$0.383{\small\pm 0.017}$ | 0.061| 6.975| | |
> | InvODE|14.000 | $1.657e^{-5}$ |$-0.025{\small\pm 0.000}$ | 0.018| - | | |
> | RoS-diffuser (ours)|0.010 | 0.010 |$1.519{\small\pm 0.330}$ | 0.106| 4.584|$100\%$ | $100\%$|
> | RoS-diffuser-CF (ours)|0.010 | 0.010 |$1.536{\small\pm 0.306}$ | ${\color{purple}0.007}$| 4.481|$100\%$ | $100\%$|
> | ReS-diffuser (ours)|0.010 | 0.010 |$1.557{\small\pm 0.289}$ | 0.107| 4.434|$46\%$ | $17\%$|
> | ReS-diffuser-CF (ours)|0.010 | 0.010 |$1.544{\small\pm 0.280}$ | ${\color{purple}0.007}$| 4.619|$36\%$ | $16\%$|
> | TVS-diffuser (ours)|0.003 | 0.003 |$1.543{\small\pm 0.303}$ | 0.107| 4.533|$47\%$ | $21\%$|
> | TVS-diffuser-CF (ours)|0.003 | 0.003 |$1.588{\small\pm 0.231}$ | ${\color{purple}0.007}$| 4.462|$48\%$ | $18\%$|
> | ReS-diffuser-l10 (ours)|0.010 | 0.010 |$1.527{\small\pm 0.291}$ | 0.011| 4.571|$39\%$ | $8\%$|
>
>
> Locomotion (CF is short for closed-form solution):
> |Experiment|  method |  S-spec($\geq0$)  | C-spec($\geq0$) | Score ($\uparrow$)  | Time($\downarrow$) |
> |---|---|---|---|---|---|
> | | Diffuser |-9.375 | -4.891 |$0.346{\small\pm 0.106}$ | ${\color{red}0.037}$|
> | | Trunc |0.0 | $\times$ |$0.286{\small\pm 0.180}$ | 0.105|
> |Walker2D| CG |-0.575 | -0.326 |$0.208{\small\pm 0.140}$ | 0.053|
> | | RoS-diffuser (ours)|0.000 | 0.010 |$0.312{\small\pm 0.165}$ | 0.183|
> | | RoS-diffuser-CF (ours)|0.000 | 0.010 |$0.321{\small\pm 0.119}$ | ${\color{purple}0.040}$|
> |---|---|---|---|---|---|
> | | Diffuser |-2.180 | -1.862 |$0.455{\small\pm 0.038}$ | ${\color{red}0.038}$|
> | | Trunc |0.0 | $\times$ |$0.436{\small\pm 0.067}$ | 0.046|
> |Hopper| CG |-0.894 | -0.524 |$0.478{\small\pm 0.038}$ | 0.047|
> | | RoS-diffuser (ours)|0.000 | 0.010 |$0.430{\small\pm 0.040}$ | 0.170|
> | | RoS-diffuser-CF (ours)|0.000 | 0.010 |$0.464{\small\pm 0.028}$ | ${\color{purple}0.040}$|
>
> Manipulation (CF is short for closed-form solution):
> |  method |  S-spec($\geq0$)  | C-spec($\geq0$) | Score ($\uparrow$)  | Time($\downarrow$) |
> |---|---|---|---|---|
> | Diffuser |-0.057 | -0.065 |$0.650{\small\pm 0.107}$ | ${\color{red}0.038}$|
> | Trunc |$1.631e^{-8}$ | $\times$ |$0.575{\small\pm 0.112}$ | 0.069|
> | CG |-0.050 | -0.053 |$0.800{\small\pm 0.328}$ | 0.075|
> | RoS-diffuser (ours)|0.072 | 0.069 |$0.925{\small\pm 0.107}$ | 0.088|
> | RoS-diffuser-CF (ours)|0.093 | 0.002 |$0.800{\small\pm 0.114}$ | ${\color{purple}0.039}$|

---

> > ### Author Response · Authors · 2024-11-18
> > **Response to review comments part II**
> >
> > **Question 1:** The forward invariance in control theory (equation 2) assumes that the control system is governed by Lipschitz functions. Is any such requirement not required for extending the definition to diffusion neural networks?
> >
> > **Response:**  This is a good point. We also need to have such a requirement for the diffusion neural networks. In fact, since the system dynamics are implicitly integrated into the training data, we would expect that the diffusion dynamics may also satisfy the assumption. Furthermore, if we consider the stochastic differential equations as the model for diffusion (as in [1]), the diffusion model would be more natural to be comparable to a control system.
> >
> > [1] Yang Song, et.al. Score-based generative modeling through stochastic differential equations. arXiv preprint arXiv:2011.13456, 2020.
> >
> >
> > **Question 2:** Diffusion dynamics is estimated online. How does error in this estimation affect the guarantees and performance in practice?
> >
> > **Response:**  This error won’t affect the guarantees for the safety of generated trajectories using diffusion models. However,  this error may actually affect the guarantees when we apply this to real systems in practice. This can be addressed using either replanning [1] or event-triggered CBF methods to catch the error bounds (defined by users) and trigger the replanning, and then we can formulate robust CBF conditions to address such an issue [2]. This definitely deserves further study.
> >
> > [1] Zhou, Siyuan, et al. "Adaptive online replanning with diffusion models." Advances in Neural Information Processing Systems 36 (2024).
> >
> > [2]  Sabouni, Ehsan, et al. "Optimal control of connected automated vehicles with event/self-triggered control barrier functions." Automatica 162 (2024): 111530.
> >
> >
> > **Question 3:** In theorem 3.2 what is the intuitive meaning of the robust term?
> >
> > **Response:**  The intuitive meaning of the robust term is that it is actually used to define a more strict safety specification compared to the original safety constraint. This is to overcome the asymptotic convergence properties of the CBF method.
> >
> > **Question 4:** If would be interesting to continually retrain the diffusion models on the generated safe data. Can this improve performance or speed up safediffuser?
> >
> > **Response:**  This is indeed a great idea. Continually retraining the diffusion models on the generated safe data can potentially improve the performance. This may be useful when we have new configurations of (static) obstacles. We would be interested in further exploring this idea in future work. As for speeding up the safediffuser, we fully addressed the issue using the proposed closed-form solutions (please see the three tables from new experiments, in response to Weakness 2).

---

> > > ### Comment · Reviewer_vnZo · 2024-11-22
> > > **Response to authors.**
> > >
> > > I would like a clarification on, "In fact, since the system dynamics are implicitly integrated into the training data, we would expect that the diffusion dynamics may also satisfy the assumption. "
> > >
> > > Is it fair to say that you "assuming" the diffusion model is  Lipschitz because it has been trained on Lipschitz dynamics? That may very well not be true and is perhaps difficult to guarantee.

---

> > > > ### Author Response · Authors · 2024-11-22
> > > > **Thanks a lot for the response**
> > > >
> > > > Yes, we assume the diffusion model is Lipschitz. We agree that this may not be true even if the data comes from dynamics that are Lipschitz. The Lipschitz requirement could be some other constraints we can add to the diffusion model, and this is something we would be interested in exploring in our future work. On the other hand, we may use exiting techniques [1] [2] [3] to train some Lipschitz diffusion models.
> > > >
> > > > [1] Fazlyab, Mahyar, et al. "Efficient and accurate estimation of lipschitz constants for deep neural networks." Advances in neural information processing systems 32 (2019).
> > > >
> > > > [2] Virmaux, Aladin, and Kevin Scaman. "Lipschitz regularity of deep neural networks: analysis and efficient estimation." Advances in Neural Information Processing Systems 31 (2018).
> > > >
> > > > [3] Pauli, Patricia, et al. "Training robust neural networks using Lipschitz bounds." IEEE Control Systems Letters 6 (2021): 121-126.

---

> > > > > ### Comment · Reviewer_vnZo · 2024-11-23
> > > > > **Response to authors.**
> > > > >
> > > > > This is an important assumption (and limitation) and it was not clear at all while reading the paper. Please clarify this in your paper. I am lowering my score to 6 in this light.

---

> ### Author Response · Authors · 2024-11-23
> **Response to the reviewer**
>
> We want to further clarify that this is not a limitation for our current methods in diffusion models. First, the diffusion models we used are in discrete time (not in continuous time) [1], and the output of the model **cannot be unbounded at each diffusion step**. Thus, the (discrete-time) diffusion models are Lipschitz.
>
> Second, even for continuous time diffusion models (such as those based on stochastic ODEs [2]), **if the activation functions are Lipschitz, the diffusion models are also Lipschitz by construction**.
>
> [1] Michael Janner, et.al., Planning with diffusion for flexible behavior synthesis. In International Conference on Machine Learning, pp. 9902–9915. PMLR, 2022
>
> [2] Yang Song, et.al., Score-based generative modeling through stochastic differential equations. arXiv preprint arXiv:2011.13456, 2020

---

> > ### Comment · Reviewer_vnZo · 2024-11-23
> > **Response to authors.**
> >
> > Thank you for the clarification. I will take a close look at the rebuttal and the revision before finalizing my recommendation.

---

> > > ### Author Response · Authors · 2024-11-23
> > > **Response to the reviewer**
> > >
> > > Thank you very much for taking the time into reconsidering your recommendation and for reviewing our paper. Please let us know if you have any other concerns.
> > >
> > > We want to emphasize here again that we have completely addressed the computational issue (mentioned as **the weakness 2 by the reviewer and reviewer hg4A**) with the new proposed closed-form solution of SafeDiffusers (given in **Appendix section B of the revised manuscript**).  We have also rerun all the experiments to show the effectiveness of the closed-form SafeDiffusers (**see the above three tables**).

---

> > > ### Author Response · Authors · 2024-11-25
> > > **Please let us know your final recommendation**
> > >
> > > Dear reviewer vnZo,
> > >
> > > Since we have addressed your concerns, and we have added new experiments of closed-form SafeDiffusers to fully address your main weakness point, would you kindly consider reevaluating your recommendation and raising the score?
> > >
> > > Please let us know if you still have any concerns,
> > >
> > > Looking forward to your reply,
> > >
> > > Best regards,
> > >
> > > Authors.

---

> > > > ### Author Response · Authors · 2024-11-29
> > > >
> > > > Dear Reviewer vnZo:
> > > >
> > > > We are truly grateful for your insightful comments and advice, which have played a significant role in enhancing the quality and clarity of our paper.
> > > >
> > > > We hope that the additional details and experimental results we provided have effectively addressed your concerns. As the rebuttal period comes to an end, we kindly request your thoughts on our rebuttal and ask that you consider raising your score accordingly. If there are any remaining concerns, please feel free to share them with us.
> > > >
> > > > Once again, we deeply appreciate your thoughtful review and constructive feedback.
> > > >
> > > > Best,
> > > >
> > > > Authors

---

> > > > > ### Comment · Reviewer_vnZo · 2024-12-01
> > > > > **Final recommendation.**
> > > > >
> > > > > Dear authors
> > > > > Thank you for actively engaging in the revision process. I am happy to increase my score to 8!

---

> > > > > > ### Author Response · Authors · 2024-12-02
> > > > > > **Thanks a lot for raising the score**
> > > > > >
> > > > > > We really appreciate the reviewer for the final recommendation and for raising the score. Thank you again for your time and effort in reviewing our paper and in the discussion.

---

### Official Review · Reviewer_hg4A · 2024-11-04

**Soundness:** 3
**Presentation:** 4
**Contribution:** 3
**Rating:** 8
**Confidence:** 4

**Summary:**

The authors present SafeDiffuser, a modified version of the Diffuser model proposed by Janner et al. (2022), adapted to ensure the satisfaction of safety constraints. They introduces "Finite-Time Diffusion Invariance," leveraging Control Barrier Functions (CBFs) within the denoising diffusion process to enforce specified safety constraints. The authors propose three approaches for implementing their methodology—ROS, RES, and TVS—that demonstrate competitive performance compared to baseline methods, achieving significantly fewer constraint violations. Experiments were conducted across Maze2D-large, locomotion, and manipulation tasks, validating the effectiveness of the proposed approaches in diverse settings.

**Strengths:**

1. The proposed method introduces a novel formulation of Control Barrier Functions (CBFs) within diffuser-family models, ensuring that safety constraints are satisfied throughout the diffusion denoising process.
2. The authors provide a systematic approach for selecting the appropriate variant based on the safety constraints.
3. Results demonstrate improvements in safety-related metrics while maintaining comparable performance in the downstream task.
4. The method can be trained using data that violates safety constraints, yet during testing, it reliably generates safe trajectories.

**Weaknesses:**

1. The proposed methods require solving Quadratic Programs (QPs) at each diffusion step, which is computationally inefficient.
2. [Minor Weakness] The approach requires predefined safety constraints. However, I think that this limitation is relatively minor, such a setup is not a major weakness.
3. It would strengthen the paper if the authors included comparisons with other methods, such as CDT [1], which also employ sequential modeling techniques to address offline RL problems with safety constraints.

[1] Liu, Zuxin, et al. "Constrained decision transformer for offline safe reinforcement learning." *International Conference on Machine Learning*. PMLR, 2023.

Despite the weaknesses, I find the formulation and results are enough to have the paper above the acceptance threshold.

**Questions:**

1. In terms of data and experimental results, could you clarify the rationale for selecting D4RL over [2], which is specifically designed for Safe Offline RL?
2. Can you clarify the motivation of applying CBFs in the diffusion denoising process steps instead of trajectory planning steps? It was not very clear.
3. Why were the experiments conducted solely in Maze2D-large? Could you clarify the choice of tasks for the locomotion experiments? Specifically, why was only the Medium-Expert variant used for the locomotion environment? What about Cheetah?
4. Could you elaborate on the generalization capabilities of the model when encountering new safety constraints beyond those specified during training? It would be valuable to understand how well the approach adapts to previously unseen constraints.


[2] Liu, Zuxin, et al. "Datasets and benchmarks for offline safe reinforcement learning." arXiv preprint arXiv:2306.09303 (2023).

---

> ### Author Response · Authors · 2024-11-18
> **Response to review comments part I**
>
> We thank the reviewer for the positive and insightful comments. We address the remaining concerns below.
>
> **Weakness 1:** The proposed methods require solving QPs, which is computationally inefficient.
>
>
> **Response:** We have fully addressed the computational issue by finding their closed-form solutions (referring to CH3 of the book ``Optimization by Vector Space Methods’’). The closed-form solution is given in Appendix section B of the revised manuscript. We have rerun all the experiments using the closed-form solution of safediffusers, which shows that the computation time is almost the same as the original diffuser (given in the following tables).
>
> Maze planning (CF is short for closed-form solution):
> |  method |  S-spec($\geq0$)  | C-spec($\geq0$) | Score ($\uparrow$)  | Time ($\downarrow$) | NLL|  Trap rate 1| Trap rate 2|
> |---|---|---|---|---|---|---|---|
> | Diffuser |-0.983 | -0.894 |$1.598{\small\pm 0.174}$ | ${\color{red}0.006}$| 4.501| | |
> | Trunc |$-1.192e^{-7}$ | -0.759 |$1.577{\small\pm 0.242}$ | 0.024| 4.494| | |
> | CG |-0.789 | -0.979 |$0.384{\small\pm 0.020}$ | 0.053| 6.962| | |
> | CG-$\epsilon$ |-0.853 | -0.995 |$0.383{\small\pm 0.017}$ | 0.061| 6.975| | |
> | InvODE|14.000 | $1.657e^{-5}$ |$-0.025{\small\pm 0.000}$ | 0.018| - | | |
> | RoS-diffuser (ours)|0.010 | 0.010 |$1.519{\small\pm 0.330}$ | 0.106| 4.584|$100\%$ | $100\%$|
> | RoS-diffuser-CF (ours)|0.010 | 0.010 |$1.536{\small\pm 0.306}$ | ${\color{purple}0.007}$| 4.481|$100\%$ | $100\%$|
> | ReS-diffuser (ours)|0.010 | 0.010 |$1.557{\small\pm 0.289}$ | 0.107| 4.434|$46\%$ | $17\%$|
> | ReS-diffuser-CF (ours)|0.010 | 0.010 |$1.544{\small\pm 0.280}$ | ${\color{purple}0.007}$| 4.619|$36\%$ | $16\%$|
> | TVS-diffuser (ours)|0.003 | 0.003 |$1.543{\small\pm 0.303}$ | 0.107| 4.533|$47\%$ | $21\%$|
> | TVS-diffuser-CF (ours)|0.003 | 0.003 |$1.588{\small\pm 0.231}$ | ${\color{purple}0.007}$| 4.462|$48\%$ | $18\%$|
> | ReS-diffuser-l10 (ours)|0.010 | 0.010 |$1.527{\small\pm 0.291}$ | 0.011| 4.571|$39\%$ | $8\%$|
>
>
> Locomotion (CF is short for closed-form solution):
> |Experiment|  method |  S-spec($\geq0$)  | C-spec($\geq0$) | Score ($\uparrow$)  | Time($\downarrow$) |
> |---|---|---|---|---|---|
> | | Diffuser |-9.375 | -4.891 |$0.346{\small\pm 0.106}$ | ${\color{red}0.037}$|
> | | Trunc |0.0 | $\times$ |$0.286{\small\pm 0.180}$ | 0.105|
> |Walker2D| CG |-0.575 | -0.326 |$0.208{\small\pm 0.140}$ | 0.053|
> | | RoS-diffuser (ours)|0.000 | 0.010 |$0.312{\small\pm 0.165}$ | 0.183|
> | | RoS-diffuser-CF (ours)|0.000 | 0.010 |$0.321{\small\pm 0.119}$ | ${\color{purple}0.040}$|
> |---|---|---|---|---|---|
> | | Diffuser |-2.180 | -1.862 |$0.455{\small\pm 0.038}$ | ${\color{red}0.038}$|
> | | Trunc |0.0 | $\times$ |$0.436{\small\pm 0.067}$ | 0.046|
> |Hopper| CG |-0.894 | -0.524 |$0.478{\small\pm 0.038}$ | 0.047|
> | | RoS-diffuser (ours)|0.000 | 0.010 |$0.430{\small\pm 0.040}$ | 0.170|
> | | RoS-diffuser-CF (ours)|0.000 | 0.010 |$0.464{\small\pm 0.028}$ | ${\color{purple}0.040}$|
>
> Manipulation (CF is short for closed-form solution):
> |  method |  S-spec($\geq0$)  | C-spec($\geq0$) | Score ($\uparrow$)  | Time($\downarrow$) |
> |---|---|---|---|---|
> | Diffuser |-0.057 | -0.065 |$0.650{\small\pm 0.107}$ | ${\color{red}0.038}$|
> | Trunc |$1.631e^{-8}$ | $\times$ |$0.575{\small\pm 0.112}$ | 0.069|
> | CG |-0.050 | -0.053 |$0.800{\small\pm 0.328}$ | 0.075|
> | RoS-diffuser (ours)|0.072 | 0.069 |$0.925{\small\pm 0.107}$ | 0.088|
> | RoS-diffuser-CF (ours)|0.093 | 0.002 |$0.800{\small\pm 0.114}$ | ${\color{purple}0.039}$|
>
> **Weakness 2:** [Minor Weakness] The approach requires predefined safety constraints. However, I think that this limitation is relatively minor, such a setup is not a major weakness.
>
> **Response:** We agree on this weakness, and have also discussed this as a limitation. However, we may learn the safety constraints using the methods such as in [1], and combine it with the diffuser. This is an ongoing research direction.
>
> [1] Robey, Alexander, et al. "Learning control barrier functions from expert demonstrations."  IEEE-CDC 2020.
>
> **Weakness 3:** It would strengthen the paper if the authors included comparisons with CDT [1], which also employ sequential modeling techniques to address offline safe RL problems.
>
> **Response:** We thank the reviewer for suggesting this. In fact, we have compared with similar methods in our tables (CG and InvODE). The method in CDT [1] can improve the safety, however, it cannot guarantee the safety as safety constraints are additional inputs to the model. The CDT [1] method is similar to that of [2] [3] (discussed in our related works), and we added it and discussed it in our revision.
>
> [1] Liu, Zuxin, et al. "Constrained decision transformer for offline safe reinforcement learning." ICML 2023.
>
> [2] Ye Yuan, et.al. Physdiff: Physics-guided human motion diffusion model
>
> [3] Anurag Ajay, et.al. Is conditional generative modeling all you need for decision making? ICLR 2023.

---

> > ### Author Response · Authors · 2024-11-18
> > **Response to review comments part II**
> >
> > **Question 1:** In terms of data and experimental results, could you clarify the rationale for selecting D4RL over [2], which is specifically designed for Safe Offline RL?
> >
> > **Response:** We totally agree that [2] is a great dataset and benchmark for safe offline RL, and we would be interested in testing our safediffusers in [2]. However, we are using D4RL since we mainly want to compare with diffusion-based methods [3], and the diffusers are implemented in D4RL. It is more fair to compare with examples in D4RL where the original diffuser [3] relied on.
> >
> > [2] Liu, Zuxin, et al. "Datasets and benchmarks for offline safe reinforcement learning." arXiv preprint arXiv:2306.09303 (2023).
> >
> > [3] Michael Janner, et.al. Planning with diffusion for flexible behavior synthesis. In International Conference on Machine Learning, pp. 9902–9915. PMLR, 2022
> >
> >
> > **Question 2:**  Can you clarify the motivation of applying CBFs in the diffusion denoising process steps instead of trajectory planning steps? It was not very clear.
> >
> > **Response:** We can apply CBFs to both the diffusion process steps and trajectory planning steps. In our case, since the trajectory is generated via the diffusion process, it is more rational to apply CBFs there. The traditional use of CBFs is in the trajectory planning steps as we combine the system dynamics with safety specifications using CBFs to guarantee safety. However, the system dynamics are implicitly incorporated into the training data, and then learned by the diffusion model, so we may just apply CBFs in the diffusion process (dynamics).
> >
> >
> > **Question 3:** Why were the experiments conducted solely in Maze2D-large? Could you clarify the choice of tasks for the locomotion experiments? Specifically, why was only the Medium-Expert variant used for the locomotion environment? What about Cheetah?
> >
> > **Response:** We have also done ablation studies on Maze2D-Umaze in which case the training data (trajectories) satisfy all the safety specifications, and this is given in Appendix B.1: ablation study 1. We chose the walker2d and hopper in our experiments as the dimension of trajectory point is lower compared to the Cheetah, and the computational complexity is lower using our methods. Now that we have given the closed-form solutions of our safediffusers, the computation issue is no longer a concern (shown in the above three new tables that the computation time is almost the same as that of diffusers). We can apply our safediffusers to any other complex systems, such as the Cheetah. Our methods can work on any task as long as we have the training data.
> >
> > **Question 4:** Could you elaborate on the generalization capabilities of the model when encountering new safety constraints beyond those specified during training? It would be valuable to understand how well the approach adapts to previously unseen constraints.
> >
> > **Response:** The generalization capabilities totally depend on the diffusion models themselves. As long as the diffusion models are trained on data that covers all the safe space (or even including unsafe space), our methods can still work well when encountering new safety constraints. This means that the diffusion model should be trained on dataset that covers the space after we add new safety constraints, and our approaches can be adapted to any unseen constraints in such cases.  We may define new unsafe sets (new safety constraints) based on the space that the training data covers, which is especially helpful in dynamical environments (such as in driving) where obstacles are always moving.

---

> ### Author Response · Authors · 2024-11-25
> **Encourage in discussion**
>
> Dear reviewer hg4A,
>
> We have added new experiments of the closed-form solution to address your main concerns. Could you please let us know if you have any other questions or concerns?
>
> Looking forward to your reply,
>
> Best regards,
>
> Authors.

---

> > ### Comment · Reviewer_hg4A · 2024-11-26
> > **Official Comment by Reviewer hg4A**
> >
> > Thank you for your clarifications and comments.
> >
> > > We have fully addressed the computational issue by finding their closed-form solutions...
> >
> > Thank you for providing these new results, I find the closed-form solution is important to improve the computational efficiency of the proposed method while maintaining the performance.
> >
> > Based on the provided results of the closed-form and potential-based diffuser (in response to reviewer gj8X), and after carefully reading the remaining comments and reviews, my primary concern has been addressed. Consequently, I have raised my score to 8.
> >
> > **Why not higher?**
> >
> > > We totally agree that [2] is a great dataset and benchmark for safe offline RL, and we would be interested in testing our safediffusers in [2]. However, we are using D4RL since we mainly want to compare with diffusion-based methods [3], and the diffusers are implemented in D4RL. It is more fair to compare with examples in D4RL where the original diffuser [3] relied on.
> >
> > I do not fully agree with this rationale. Your work introduces a new method for safe planning, which goes beyond merely comparing it to the diffuser—a method not inherently designed with safety considerations. Furthermore, training the diffuser [3] with [2] could have been easily accomplished. I still think that this comparison is important, at the very least, for the benefit of the broader community. I agree with you that it would not be fair to provide the results only in [2], but providing the results in D4RL and [2] would make it complete.
> >
> > [2] Liu, Zuxin, et al. "Datasets and benchmarks for offline safe reinforcement learning." arXiv preprint arXiv:2306.09303 (2023).
> >
> > [3] Michael Janner, et.al. Planning with diffusion for flexible behavior synthesis. In International Conference on Machine Learning, pp. 9902–9915. PMLR, 2022

---

> > > ### Author Response · Authors · 2024-11-26
> > > **Response to reviewer**
> > >
> > > Thanks a lot for raising the score to 8.  Training diffusers [3] on more dataset and benchmarks [2] would definitely make broader impact for the work. We would consider including those datasets and benchmarks in the follow-up work.
> > >
> > > Please do not hesitate to let us know if you have any other comments/suggestions.

---

### Official Review · Reviewer_4UNS · 2024-11-07

**Soundness:** 2
**Presentation:** 2
**Contribution:** 2
**Rating:** 6
**Confidence:** 3

**Summary:**

The authors propose a method to use control barrier functions (CBFs) with diffusion planning, enabling safety guarantees. The results are demonstrated on a 2D maze problem, Walker2D and a manipulation problem from related work.

**Strengths:**

- Safety guarantees are important for some applications and of interest to the safe planning community
- This may be the first (only?) safety method for diffusion planning.

**Weaknesses:**

- Some parts are a bit difficult to read and have unusual terminology (e.g., "local traps"?) and the paper contains some minor grammar and cosmetic issues which make the quite complex approach harder to understand.
- The method is actually three methods, and the user is to pick the one most suitable for their problem. This seems a bit complicated, why not go for the most general?

Example presentation issues (I recommend you do another polish pass):
- General: excessive underlining
- 470: typo: physical simulator
- 474: typo: can work in general safety guarantees

**Questions:**

1) Are "local traps" = local minima?
2) Isn't it a big problem that the method gets stuck in "local traps" between 39% and 100% of the time for some of the benchmarks? What does this mean in practice?
3) What is trap rate 1 vs. trap rate 2 in the table?
4) Could you comment on why you chose to use CBF here instead of e.g. lagrangian objectives common in safe RL?
5) Why three methods instead of the most general, or one that picks the best solution approach?

---

> ### Author Response · Authors · 2024-11-18
> **Response to review comments**
>
> We appreciate the reviewer for all the constructive comments. We address the remaining concerns in the response below.
>
> **Weakness 1:** Some parts are a bit difficult to read and have unusual terminology (e.g., "local traps"?) and the paper contains some minor grammar and cosmetic issues which make the quite complex approach harder to understand.
>
> **Response:** The term ``local traps’’  denote trajectory points getting stuck at obstacle boundaries and is due to the existence of non-convex unsafe sets, which are explicitly explained at the end of Sec. 1 and 2 where they firstly show up. We added a formal definition (Def. 3.2) in the revision to make it more clear. We revised the paper to address all the grammar and cosmetic issues.
>
> **Weakness 2:** The method is actually three methods, and the user is to pick the one most suitable for their problem. This seems a bit complicated, why not go for the most general?
>
> **Response:** The three methods are at different complexity and for different purposes. The RoS diffuser is simple to use but may be subject to local trap issues if there are non-convex unsafe sets, while the other two methods are more complex (e.g., designing either relaxation variables or time-varying functions) but can address the local trap issues. As far as we are concerned, there is no such a general method that can simultaneously be simple and address all the concerns.  Nevertheless, these three methods are not that different from the implementation perspective, and they are backup'ed by different theoretical justification.
>
> **Weakness 3:** presentation issues: excessive underlining and typos
>
> **Response:**  We removed unnecessary underlining and revised the paper to address typos in the revision.
>
>
> **Question 1:** Are "local traps" = local minima?
>
>
> **Response:** Conceptually, yes, it is similar to local minima. However, since this is for planning purposes, local traps denote trajectory points getting stuck at obstacle boundaries and are due to the existence of non-convex unsafe sets (may not be necessarily a minima, but more like a trap). Please see Fig. 6 of appendix for an example of local traps in maze planning.
>
> **Question 2:** Isn't it a big problem that the method gets stuck in "local traps" between 39% and 100% of the time for some of the benchmarks? What does this mean in practice?.
>
> **Response:** This is not a big problem **if only few** trajectory points get struck in local traps (as in our proposed ReS and TVS diffusers) as we can easily address this using smoothing techniques (e.g., by removing these trajectory points or taking the average of trajectory points within a sliding window). In practice, this means that some trajectory points get stuck at unsafe set boundaries (usually flat or nonconvex). Please see Fig. 6 of appendix for an example of local traps in maze planning.
>
> **Question 3:** What is trap rate 1 vs. trap rate 2 in the table?
>
> **Response:** The trap rate $r$ denotes the trap rate with the number of trapped trajectory points $\geq r$, where $r=1$ or $2$ in the table. This was defined in the caption (the last sentence) of Table 2 (highlighted in the revision).
>
>
> **Question 4:**  Could you comment on why you chose to use CBF here instead of e.g. lagrangian objectives common in safe RL?
>
> **Response:** The Lagrangian objectives in safe RL can also guarantee safety. However, since it is very complex to solve, to our knowledge, it is mostly used as a loss/reward function during the training of safe RL. There are still no safety guarantees during inference. We choose CBFs since the CBF method can strictly ensure the safety (as the CBF constraints are hard constraints) while being very efficient to solve during both training and inference, as shown in Thm. 3.2-3.4. We can also find the closed-form solution of the CBF method (please see **Appendix section B of the revised manuscript** for details). We have done new experiments to show the effectiveness of the closed-form solution, please see **the response to weakness 1 of reviewer hg4A** for the new experiment results.
>
> **Question 5:** Why three methods instead of the most general, or one that picks the best solution approach?
>
> **Response:** As far as we are concerned, there is no such a general method that can simultaneously be simple and address all the concerns.  The RoS diffuser is simple to use but may be subject to local trap issues if there are non-convex unsafe sets, while the other two methods are more complex (e.g., designing either relaxation variables or time-varying functions) but can address the local trap issues. Nevertheless, these three methods are not that different from the implementation perspective, and they are backup'ed by different theoretical justification.

---

> > ### Comment · Reviewer_4UNS · 2024-11-25
> > **Response to authors**
> >
> > I thank the authors for the informative and candid answers. I still have some concerns about the complexity of using this approach for a practitioner, as well as its limitations w.r.t. "local traps".
> >
> > Fig.6 in the appendix was very illuminating. If I were to compare this approach to e.g. classical optimal control / trajectory planning approaches, I would say that parts of your trajectory getting stuck on the wrong side of a barrier function looks like a local minima for an infeasible-start trajectory optimizer. I'm not entirely up to date on diffusion planning, but between constrained trajectory/motion planning methods, Safe RL, or combinations of the two, I'm struggling with seeing for what class of problems that SafeDiffuser holds promise to advance state of the art. Especially considering that none of the benchmarks look particularly difficult. That said, not every paper needs to push SOTA numbers and I believe this paper contains some theory building blocks that could advance the theory of (safe) diffusion planning. I will observe the other reviewer reactions and consider increasing my score based on their responses..

---

> > > ### Author Response · Authors · 2024-11-26
> > > **Response to the reviewer**
> > >
> > > We thank the reviewer for the helpful feedback. We provide further clarifications below.
> > >
> > > **1. complexity of using this approach**
> > >
> > > **Response:** By complexity, if the reviewer refers to the computational complexity, then we have provided the closed-form solution, and conducted new experiments (see the three tables in the overall summary) to show that the computation complexity is almost the same as the original diffusion models, which shows that the computational complexity is very low.
> > >
> > > If the reviewer refers to the complexity of using diffusion for planning instead of other approaches, then we would argue that diffusion models can easily do planning in any space (e.g., state space, Cartesian space {such as in the 3D x-y-z frame}) without requiring the knowledge of the system model (e.g., robot dynamics/kinematics), while traditional motion planning methods (such as A*, RRT) usually do planning in the Cartesian space (i.e., 3D position trajectory) and require the system dynamics/kinematics in order to find the state trajectory. Therefore, **diffusion models are easier if we need to do planning in the lower level state space.**
> > >
> > > **2. limitations w.r.t. "local traps"**
> > >
> > > **Response:** The local traps happen due to the fact that the diffusion process is overly constrained (e.g., from the initial reverse diffusion time T to the final reverse diffusion time 0).  This problem can be addressed by applying our approaches to limited number of diffusion steps (e.g., the last 10 diffusion steps) in which case the generated trajectory has already passed the local traps (as shown by the ReS-diffuser-l10 in Table 2).   This problem can also be addressed using the proposed ReS-diffuser and TVS-diffusers. It is worth noting that there are also many other approaches from optimizations that can address this local trap/minima problem, such as the popular boosting methods. In other words, whenever we detect such an trapped trajectory point (using the new added Def 3.2 in the revision), we can apply some additional ``energy'' to it to make it escape from local trap/minima. Our focus of this paper is to show the formal/theoretical safety guarantees of diffusion model.
> > >
> > > **3. local trap v.s. local minima**
> > >
> > > **Response:** The local trap problem in SafeDiffusers can be taken as the local minima if we use optimization-based planning methods. However, since diffusion do not involve optimizations, we call it local trap as this may happen when the trajectory points lose the momentum to escape from local traps formed by unsafe sets.
> > >
> > > **4. what class of problems that SafeDiffuser holds promise to advance state of the art**
> > >
> > > **Response:** There are two classes of problems that SafeDiffuser holds promise to advance state of the art:
> > >
> > > (1) Planning in the state space instead of the Cartesian space (e.g., x-y-z frame).  The controls of robots are usually in the state space (e.g., joint speed of a manipulator). Existing methods usually do planning for the robot in the Cartesian space and then find the state trajectory using robot kinematics/dynamics, which is complicated. While SafeDiffusers do not require such robot kinematics/dynamics, and thus is easier (the robot dynamics are inherited in the training data).  Two of such examples are the locomotion and manipulation shown in the experiments (the planning is in the joint space of robots).
> > >
> > > (2) Scalable planning capabilities/behavior synthesis. We can easily fuse multiple diffusion models (e.g., trained on different dataset) into a single one to synthesis the robot behaviors, and to get novel behaviors. This has been extensively studied in [1].  **This shows the scalability and flexibility of diffusion-based planning methods**.
> > >
> > > [1] Michael Janner, Yilun Du, Joshua Tenenbaum, and Sergey Levine. Planning with diffusion for flexible behavior synthesis. In International Conference on Machine Learning, pp. 9902–9915. PMLR, 2022

---

> ### Author Response · Authors · 2024-11-25
> **Please let us know your feedback**
>
> Dear reviewer 4UNS,
>
> Hope our response have addressed your concerns. Could you please let us know if you have any further concerns?
>
> Looking forward to your reply,
>
> Best regards,
>
> Authors.

---

> ### Author Response · Authors · 2024-11-27
> **Could you please let us know your feedback**
>
> Dear Reviewer 4UNS,
>
> We have got responses from all the other reviewers, and they are satisfied with our current revision of the paper indicated by raising their scores.  As you have explicitly mentioned that ``I will observe the other reviewer reactions and consider increasing my score based on their responses'', could you please let us know your response as all the other reviewers have responded?
>
> Looking forward to your reply,
>
> Thank you and best regards,
>
> Authors.

---

> ### Author Response · Authors · 2024-11-29
>
> Dear Reviewer 4UNS:
>
> We are truly grateful for your insightful comments and advice, which have played a significant role in enhancing the quality and clarity of our paper.
>
> We hope that the additional details and experimental results we provided have effectively addressed your concerns. As the rebuttal period comes to an end, we kindly request your thoughts on our rebuttal and ask that you consider raising your score accordingly. If there are any remaining concerns, please feel free to share them with us.
>
> Once again, we deeply appreciate your thoughtful review and constructive feedback.
>
> Best,
>
> Authors

---

> ### Comment · Reviewer_4UNS · 2024-11-29
> **Raising score from 5 to 6.**
>
> In light of the updates and clarifications I am raising my score to 6. I think the approach shows promise but I am not entirely convinced of how practical it is to apply. The experiments are on rather simple and low-dimensional problems: 2D mazes, 2D walker and what seems like a rather simple 7DoF manipulation problem. It is not obvious to me on which problem I would prefer this over some combination of classical (e.g., MPC/MPPI, RRT*) , Safe RL or safety filter approaches. It does impose safety during run-time but you may also need special handling of parts of the trajectory getting stuck local traps, rendering the solution infeasible.

---

> > ### Author Response · Authors · 2024-11-30
> > **Thanks a lot for raising the score**
> >
> > We really appreciate the reviewer for raising the score.  The experiments follow the Diffuser paper [1] for better clarification and fair comparison.  This work focuses on the theoretical safety guarantees of diffusion models, and extensive applications (as well as figuring out the advantages of diffusion over classical approaches) deserve a separate paper.  The local trap problem can be addressed using the proposed ReS and TVS diffusers, and we may even run several safediffusers in parallel (this is possible as the closed-form solution is also now provided), and select the best trajectory among them.
> >
> > [1] Michael Janner, et.al., Planning with diffusion for flexible behavior synthesis. In International Conference on Machine Learning, pp. 9902–9915. PMLR, 2022

---

### Author Response · Authors · 2024-11-20
**Summary of our response to address reviewers' main concerns**

We appreciate all the reviewers for their constructive comments. We have added extensive experiments to fully address the limitations of our SafeDiffusers, and the paper (PDF) has also been updated accordingly.  Here we summarize our response to their main concerns.

**1. New experiments of closed-form solution to SafeDiffusers [Reviewers hg4A, vnZo, gj8X].**  We have fully addressed the computational issue by finding their closed-form solutions.  **The closed-form solution is given in Appendix section B of the revised manuscript**. We have rerun all the experiments using the closed-form solution of safediffusers, which shows that the computation time is almost the same as the original diffuser (given in the following three tables).

Maze planning (CF is short for closed-form solution):
|  method |  S-spec($\geq0$)  | C-spec($\geq0$) | Score ($\uparrow$)  | Time ($\downarrow$) | NLL|  Trap rate 1| Trap rate 2|
|---|---|---|---|---|---|---|---|
| Diffuser |-0.983 | -0.894 |$1.598{\small\pm 0.174}$ | ${\color{red}0.006}$| 4.501| | |
| Trunc |$-1.192e^{-7}$ | -0.759 |$1.577{\small\pm 0.242}$ | 0.024| 4.494| | |
| CG |-0.789 | -0.979 |$0.384{\small\pm 0.020}$ | 0.053| 6.962| | |
| CG-$\epsilon$ |-0.853 | -0.995 |$0.383{\small\pm 0.017}$ | 0.061| 6.975| | |
| InvODE|14.000 | $1.657e^{-5}$ |$-0.025{\small\pm 0.000}$ | 0.018| - | | |
| RoS-diffuser (ours)|0.010 | 0.010 |$1.519{\small\pm 0.330}$ | 0.106| 4.584|$100\%$ | $100\%$|
| RoS-diffuser-CF (ours)|0.010 | 0.010 |$1.536{\small\pm 0.306}$ | ${\color{purple}0.007}$| 4.481|$100\%$ | $100\%$|
| ReS-diffuser (ours)|0.010 | 0.010 |$1.557{\small\pm 0.289}$ | 0.107| 4.434|$46\%$ | $17\%$|
| ReS-diffuser-CF (ours)|0.010 | 0.010 |$1.544{\small\pm 0.280}$ | ${\color{purple}0.007}$| 4.619|$36\%$ | $16\%$|
| TVS-diffuser (ours)|0.003 | 0.003 |$1.543{\small\pm 0.303}$ | 0.107| 4.533|$47\%$ | $21\%$|
| TVS-diffuser-CF (ours)|0.003 | 0.003 |$1.588{\small\pm 0.231}$ | ${\color{purple}0.007}$| 4.462|$48\%$ | $18\%$|
| ReS-diffuser-l10 (ours)|0.010 | 0.010 |$1.527{\small\pm 0.291}$ | 0.011| 4.571|$39\%$ | $8\%$|


Locomotion (CF is short for closed-form solution):
|Experiment|  method |  S-spec($\geq0$)  | C-spec($\geq0$) | Score ($\uparrow$)  | Time($\downarrow$) |
|---|---|---|---|---|---|
| | Diffuser |-9.375 | -4.891 |$0.346{\small\pm 0.106}$ | ${\color{red}0.037}$|
| | Trunc |0.0 | $\times$ |$0.286{\small\pm 0.180}$ | 0.105|
|Walker2D| CG |-0.575 | -0.326 |$0.208{\small\pm 0.140}$ | 0.053|
| | RoS-diffuser (ours)|0.000 | 0.010 |$0.312{\small\pm 0.165}$ | 0.183|
| | RoS-diffuser-CF (ours)|0.000 | 0.010 |$0.321{\small\pm 0.119}$ | ${\color{purple}0.040}$|
|---|---|---|---|---|---|
| | Diffuser |-2.180 | -1.862 |$0.455{\small\pm 0.038}$ | ${\color{red}0.038}$|
| | Trunc |0.0 | $\times$ |$0.436{\small\pm 0.067}$ | 0.046|
|Hopper| CG |-0.894 | -0.524 |$0.478{\small\pm 0.038}$ | 0.047|
| | RoS-diffuser (ours)|0.000 | 0.010 |$0.430{\small\pm 0.040}$ | 0.170|
| | RoS-diffuser-CF (ours)|0.000 | 0.010 |$0.464{\small\pm 0.028}$ | ${\color{purple}0.040}$|

Manipulation (CF is short for closed-form solution):
|  method |  S-spec($\geq0$)  | C-spec($\geq0$) | Score ($\uparrow$)  | Time($\downarrow$) |
|---|---|---|---|---|
| Diffuser |-0.057 | -0.065 |$0.650{\small\pm 0.107}$ | ${\color{red}0.038}$|
| Trunc |$1.631e^{-8}$ | $\times$ |$0.575{\small\pm 0.112}$ | 0.069|
| CG |-0.050 | -0.053 |$0.800{\small\pm 0.328}$ | 0.075|
| RoS-diffuser (ours)|0.072 | 0.069 |$0.925{\small\pm 0.107}$ | 0.088|
| RoS-diffuser-CF (ours)|0.093 | 0.002 |$0.800{\small\pm 0.114}$ | ${\color{purple}0.039}$|


**2. Comparison to more baselines to explain why we propose SafeDiffusers instead of using other methods (potential-based, reachability, Lagrange-based RL) [Reviewers 4UNS, gj8X]**
We have added new experiments to compare the potential-field based method and our methods (see the table below). In summary, potential-field based/existing methods have no strict safety guarantees (see Fig. 5a in ``Luo, Y., Sun, C., Tenenbaum, J. B., & Du, Y. (2024). Potential based diffusion motion planning. arXiv preprint arXiv:2407.06169'' showing there are still collisions with potential-based methods), while our methods do have guarantees.

Maze planning:
|  method |  S-spec($\geq0$)  | C-spec($\geq0$) | Score ($\uparrow$)  | Time ($\downarrow$) | NLL|
|---|---|---|---|---|---|
| Potential-based diffuser |-0.798 | -0.908 |$0.916{\small\pm 0.556}$ | 0.015| 5.289|
| RoS-diffuser-CF (ours)|0.010 | 0.010 |$1.536{\small\pm 0.306}$ | $0.007$| 4.481|
| ReS-diffuser-CF (ours)|0.010 | 0.010 |$1.544{\small\pm 0.280}$ | $0.007$| 4.619|
| TVS-diffuser-CF (ours)|0.003 | 0.003 |$1.588{\small\pm 0.231}$ | $0.007$| 4.462|

**We hope our responses below convincingly address all the reviewers’ concerns. We thank all the reviewers for their time and efforts again.**

---

### Author Response · Authors · 2024-12-03
**Summary of Rebuttal and Paper Revision**

We thank all the reviewers and AC for reviewing our paper, and for being active in the rebuttal phase. We have revised the paper accordingly.  Here, we summarize our rebuttal and revision below:

 1. We have added the closed-form solution of SafeDiffusers (Appendix Section B of the revision) to fully address the computational concerns, and we have rerun all the experiments to show the effectiveness of the closed-form solution (reviewers hg4A and vnZo).

2. We have added a comparative study between the proposed SafeDiffusers and potential-based diffusion models, with results showing the advantage of our methods in safety guarantees (reviewers hg4A and gj8X).

3. We have added a formal definition of the local trap (Def. 3.2 in the revision) in navigation problems, which makes the concept clear (reviewer 4UNS).

4. We have fully clarified the Lipschitz problem for the diffusion model by referring to existing literatures, and revised the paper accordingly (after equation (5) of the revision) (reviewer vnZo).

5. We have added a unified and simple definition of the proposed SafeDiffusers (the general form of SafeDiffusers, as shown in Def. 3.3 of the revision), and have made connections to the three SafeDiffusers after Thms. 3.4-3.6 (reviewers 4UNS and gj8X).

All the reviewers have responded that we have addressed their concerns indicated by raising their scores in the rebuttal phase. We appreciate all the reviewers and AC again for taking the time in reviewing our paper.

---

### Meta-Review · Area_Chair_Zq4p · 2024-12-24

**Metareview:**

The paper proposes SafeDiffuser, a framework that integrates control barrier functions (CBFs) into diffusion processes to ensure the safety of diffusion-based planners. The model operates by using a quadratic program during each step of the denoising process to identify a perturbation that guides the generated trajectory towards the safety set. The paper evaluates SafeDiffuser alongside contemporary baselines in terms of task performance and adherence to safety constraints on various robot planning tasks.

The paper was reviewed by four referees, who agree on several of the paper's key strengths and weaknesses. Among its strengths, the ability to ensure that diffusion-based planners adhere to safety constraints becomes increasingly important as more and more people use diffusion models for planning. Further, at least two reviewers note the novelty in the paper's integration of CBFs within diffusion models. The experiments support SafeDiffuser's ability to ensure the safety of the planned trajectories, while maintaining performance that is comparable to traditional diffusion-based planners. However, several reviewers raised concerns about SafeDiffuser's computational complexity due to the need to solve a quadratic program at each diffusion step. Notably, the authors derived a closed-form solution of the model and re-ran all of the experiments, the result of which reveals that the computational cost is not significantly higher than that of the vanilla Diffuser model. Additionally, several reviewers called into question the difficulty of implementing SafeDiffuser due to the need to choose among the three variants of the model as well as the need to design the CBFs, which can be particularly challenging for systems with highly nonlinear dynamics. The authors addressed some of these concerns during the rebuttal period, proposing a unified version of SafeDiffuser and connected it to the three variants proposed in the original submission. Additionally, at least two reviewers questioned the validity of the assumption that the diffusion process is Lipschitz, which the authors clarified in their responses.

Overall, the authors put significant effort into revising the paper to address the reviewers' initial concerns. It is evident to the AC that this effort went a log way in clarifying the paper's contributions.

**Additional Comments On Reviewer Discussion:**

There was an extensive amount of discussion between the reviewers and the authors. This discussion, together with the effort the authors put into deriving the closed-form solutions and re-running all of the experiments, significantly improved the quality of the paper.

---

### Decision · Program_Chairs · 2025-01-22

Accept (Poster)